# BRIDGE: BI-LEVEL REINFORCEMENT LEARNING FOR DYNAMIC GROUP STRUCTURE IN COALITION FORMATION GAMES

**Shuqing Shi**[1], **Nam Phuong Tran**[2], **Hao Liang**[1], **Debmalya Mandal**[2], **Long Tran-Thanh**[2], **Yali Du**[1,3]
[1]King's College London, [2]University of Warwick, [3]The Alan Turing Institute
{shuqing.shi,hao.liang,yali.du}@kcl.ac.uk
{nam.p.tran,Debmalya.Mandal,long.tran-thanh}@warwick.ac.uk

## ABSTRACT

Coalition Formation Games investigate how a group of autonomous agents voluntarily organize into subgroups (i.e., coalitions) to achieve common goals or maximize collective utility. This field has been a subject of long-standing research within game theory and related disciplines. The core challenge in these games lies in efficiently exploring the exponentially large space of possible coalition structures to identify the optimal partition. While existing approaches to solve coalition formation games either provide exact solutions with limited scalability or approximate solutions without quality guarantees, we propose a novel scalable and sample-efficient approximation method based on deep reinforcement learning. Specifically, we model the coalition formation game as a finite Markov Decision Process (MDP) and utilize deep neural networks to approximate the optimal value functions within both the full and abstracted coalition structure spaces, thereby indirectly deriving optimal coalition structures. Furthermore, our method can be leveraged for bi-level optimization problems where coalition values are determined by the policies of individual agents at a lower decision-making level. This way, our approach can facilitate dynamic, adaptive adjustments to coalition value assessments as they evolve over time. Empirical results demonstrate our algorithm's effectiveness in approximating optimal coalition structures in both normal-form and mixed-motive Markov games.

## 1 INTRODUCTION

Coalition formation games constitute a pivotal area of research within multi-agent systems, focusing on enabling groups of agents to collaborate in accomplishing specific tasks (Shehory & Kraus, 1998). These games are crucial for optimizing collaborative efforts and resource allocation among autonomous entities, exemplified by applications like cooperative social ride-sharing (Bistaffa et al., 2017), disaster response coordination (Diehl & Adams, 2023; Mouradian et al., 2017) and smart grid management (Chiş & Koivunen, 2017; Han et al., 2019). A central challenge in this game is the *Coalition Structure Generation (CSG)* problem (Sandholm et al., 1998; Rahwan, 2008; Dang & Jennings, 2004; Aziz & de Keijzer, 2011), which requires partitioning a set of agents into mutually exclusive and collectively exhaustive coalitions, referred to as a *coalition structure*, to maximize social welfare.

Determining the optimal coalition structure poses a significant computational challenge, classified as NP-complete (Sandholm et al., 1999). Existing approaches have traditionally fallen into two categories: exact and approximation methodologies. Exact methods (Yun Yeh, 1986; Rahwan et al., 2009) guarantee optimality but are computationally prohibitive beyond small scales, leading to limitations in scalability and restricting their applicability to small-scale problems (e.g., fewer than 40 agents (Rahwan et al., 2015)). This inherent constraint renders them less viable for real-world scenarios involving a substantial number of agents. Conversely, approximation methods (Di Mauro et al., 2010; Farinelli et al., 2013) offer more computationally efficient solutions but typically lack theoretical guarantees. Critically, both of these established lines of research are primarily designed for normal-form games, necessitating re-computation for each new problem instance and lacking

mechanisms to address sequential decision-making in Markov games. Importantly, while any combinatorial problem can theoretically be encoded as an MDP, such naive formulations explode exponentially and provide no exploitable structure—computing optimal policies for arbitrary MDPs is PSPACE-hard (Littman, 1996). Our contribution lies in designing a structured, RL-tractable formulation with compact representations and interpretable transitions.

In this work, to address the limitations of traditional CSG approaches, we formulate the CSG problem within the framework of a Markov Decision Process (MDP). To address the complex, sequential decision-making inherent in Markov games, we propose a novel bi-level reinforcement learning framework. Our framework is designed with an upper level focused on optimizing coalition formation, modeled as an episodic MDP where states represent coalition structures and actions involve merging coalitions. The reward at this upper level is defined by the change in coalition structure value, a crucial design choice that enables the generalization of learned values to unseen structures by exploiting the compositional nature of coalition values derived from their constituent sub-coalitions.. Simultaneously, the lower level of our framework addresses the optimization of individual agents strategies within the coalition structure dictated by the upper level, with agents learning their best responses to the current coalition structure. This bi-level architecture facilitates a dynamic adaptation of coalition structures in response to the learning strategies of the lower-level agents, leading to potentially more robust and flexible collective behavior in dynamic multi-agent environments. The use of deep neural networks for function approximation in both levels allows our method to achieve efficient inference for approximating coalition structure values and guiding individual agent policy determination, ultimately leading to optimal coalition structures.

Our contributions can be summarized as follows: First, a novel MDP formulation with structure-consistent $N \times N$ state representation, $O(N)$-dimensional action encoding via shared scorer $\psi$, and difference-based rewards enabling generalization; Second, BRIDGE, a bi-level RL framework with measurable equilibrium selection $\sigma : \text{NE}(s_l) \to \Pi_f$ ensuring deterministic transitions; Third, comprehensive experiments showing generalization from 3 to 100 agents, faster inference than traditional coalition structure generation baselines at scale, and robustness to follower suboptimality.

## 2 RELATED WORK

The CSG problem has been addressed by a variety of algorithms, ranging from exact to approximate solutions. Exact methods, such as Integer Programming (IP) (Rahwan et al., 2009), guarantee optimality but suffer from high computational complexity. Notable examples include ODP-IP (Changder et al., 2019), which employs imperfect dynamic programming, and ODSS (Changder et al., 2020), which optimizes the search space by integrating IP with Integer Dynamic Programming (IDP) (Rahwan, 2008). Hybrid exact algorithms (Michalak et al., 2016) for complete set partitioning, have also been adapted for CSG. However, the inherent combinatorial nature of CSG limits the scalability of these exact approaches. In contrast, approximate algorithms aim to find high-quality solutions within reasonable time. Fast code based algorithm (FACS) (Taguelmimt et al., 2021) utilizes a specific search space representation and heuristics for efficient exploration. The Parallel Index-based Search Algorithm (PICS) (Taguelmimt et al., 2022) offers an anytime and more scalable solution for larger problems. Our paper also contributes an approximation method based on deep reinforcement learning to tackle large-scale CSG instances.

Furthermore, the principles of CSG find natural extensions to the domain of Markov games, particularly in the context of multi-agent systems. Concepts such as task allocation (Zhao et al., 2019; Iqbal et al., 2022; Chen et al., 2017; Peng et al., 2017; Shu & Tian, 2018) and group division (Lhaksmana et al., 2018; Macarthur et al., 2011; Russell & Zimdars, 2003; Schneider, 1999), widely studied within multi-agent reinforcement learning (MARL), exhibit a fundamental resemblance to the challenges of forming effective coalitions in mixed-motive Markov games(Li et al., 2024). This inherent similarity provides a strong motivation for adopting a bi-level reinforcement learning approach in this work to address the complexities of CSG in such settings.

## 3 PRELIMINARIES AND DEFINITIONS

In this section we first introduce the classic CSG setting then propose its extension to Bi-level Markov Games.

**Coalition Structure Generation**  A CSG problem is of size $n$ if it is defined on a set of $n$ agents $\mathcal{N} := \{1, 2, \ldots, n\}$. A coalition $C$ is any non-empty subset of $\mathcal{N}$. The characteristic function $v : 2^{\mathcal{N}} \to \mathbb{R}$ assigns a real value to each coalition, indicating the value that this coalition could obtain if they cooperated. A coalition structure $CS$ is a partition of the agent set $\mathcal{N}$ into disjoint coalitions. Formally, a coalition structure $CS$ consists of a set of non-empty coalitions $CS = \{C_1, C_2, \ldots, C_k\}$, where $k = |CS|$ is the cardinality of $CS$. $CS$ satisfies the following properties: $\bigcup_{j=1}^{k} C_j = \mathcal{N}, C_i \cap C_j = \emptyset, \forall i \neq j \in \{1, 2, \ldots, k\}$. The value of $CS$ is $v(CS) := \sum_{C \in CS} v(C)$. The goal is to find the optimal $CS^*$ that maximizes the value of coalition structure:

$$CS^* := \arg \max_{CS \in \Pi(\mathcal{N})} v(CS),$$

where $\Pi(\mathcal{N})$ denotes the set of coalition structures on $\mathcal{N}$. The problem of searching for the optimal coalition structure is NP-complete.

**Bi-level Markov Games**  Following the definition of the Stackelberg Game (Simaan & Cruz Jr, 1973), we define the high-level agent as the *leader agent* and the low-level agents as the *follower agents*. The goal of the *leader* agent is to obtain the optimal coalition structure. To achieve this, we define the leader's search process as an episodic MDP $\mathcal{M}_l = \langle S_l, \mathcal{T}_l, A_l, r_l, H_l \rangle$. Here, $S_l = \Pi(\mathcal{N})$ is the set of coalition structures. For each time step $t_c \in \{1, 2, \cdots, H_l\}$, $s_{l,t_c} = \{C_1, \ldots, C_k\} \in S_l$ represents a coalition structure at time step $t_c$. The initial state is the singleton partition $s_{l,0} := \{\{1\}, \ldots, \{N\}\}$. An action $a_l$ corresponds to choosing any two coalitions given $s_l$ to merge, that is, $a_l \in \{(C_i, C_j) \mid 0 \leq i < j \leq |s_l|\} \cup \varnothing$ (where the action $a_l = \varnothing$ keeps the coalition structure unchanged). Next, given the current state $s_{l,t_c}$ at time step $t_c$ and action $a_{l,t_c}$, we define a deterministic transition function $\mathcal{T}_l$ as: $\mathcal{T}(s_{l,t_c}, a_{l,t_c}) := (s_{l,t_c} \setminus a_{l,t_c}) \cup (\bigcup_{C \in a_{l,t_c}} C)$. We set the horizon $H_l = N - 1$. The horizon $H_l = N - 1$ is a natural upper bound: each merge reduces coalition count by one, so at most $N - 1$ merges reach the grand coalition. The null action enables early termination for non-grand-coalition outputs. The reward of leader $r_l$ is defined as the difference between the next coalition structure value of the followers and that of the current step:

$$r_l(s_l, a_l, \pi_f) := \sum_{C \in T(s_l, a_l)} J_f^C(\pi_f \mid s_l) - \sum_{C \in s_l} J_f^C(\pi_f \mid s_l), \tag{1}$$

where the follower agents' policies are defined as $\pi_f : S_f \times S_l \to \Delta(A_f)$, the follower agents' coalition cumulative reward under such policies is defined as $J_f^C(\pi_f)$. Given a (stochastic) leader's policy $\pi_l$, we define the expected reward of the leader as

$$J_l(\pi_l, \pi_f) := \mathbb{E}\left[\sum_{t_c=0}^{N-1} r_{l,t_c}(s_{l,t_c}, a_{l,t_c}, \pi_f) \mid \pi_l\right].$$

For the low-level cooperative game, we define the cooperative game as $\mathcal{M}_f := \langle N, S_f, \mathcal{T}_f, A_f, r_f, CS, \gamma_f \rangle$(Tran et al., 2024; Du et al., 2023). The environment state is given by $s_{f,t_k} \in S_f$. At each episode $k$ at each timestep $t_k$, each low-level agent $i \in N$ receives state $s_{f,t_k}^i$ and chooses an action $a_{f,t_k}^i \sim \pi_f^i$. The instantaneous reward is defined as real-valued functions conditioned on the coalition: $r_f^C : \mathcal{S}_f \times \mathcal{A}_f \to \mathbb{R}$. The actions of all $N$ agents form a joint action $\boldsymbol{a}_{f,t_k} := [a_{f,t_k}^1, \cdots, a_{f,t_k}^N]$. The states evolves according to the transition function $\mathcal{T}_f(s_{f,t_k+1} \mid \boldsymbol{a}_{f,t_k}, s_{f,t_k})$. Each follower agent $i$ seeks to maximize the expected discounted rewards of its coalition, defined by

$$J_f^C(\pi_f) := \sum_{t_k=0}^{\infty} \mathbb{E}\left[\gamma_f^{t_k} r_{f,t_k}^C(s_{f,t_k}, a_{f,t_k}) \mid \pi_f\right],$$

where $\gamma_f \in (0, 1)$ is the discount factor and $r_{f,t_k}^C(s_f, a_f)$ is reward for coalition $C \in CS$. Due to the non-additive structure of the reward function (e.g., when collaboration among agents might incur some cost), the grand coalition $\mathcal{N}$ might not be the optimal one.

## 4 METHODOLOGY

### 4.1 MDP FORMULATION FOR CSG

The conventional CSG problem centers on the identification of a coalition structure that optimizes social welfare. This optimization objective can be reformulated from the perspective of a single leader agent.

In this work, we employ a deep reinforcement learning methodology to approximate the value of the optimal coalition structure. We define the state of the leader agent, denoted as $s_l$, as the current coalition structure. The action space of the leader agent, $a_l$, consists of decisions regarding which pair of existing coalitions within the current structure should be merged. The reward function for the leader agent, $r_l$, is defined as the difference in the value of the coalition structure between the subsequent state, $s_{l,t_c+1}$, and the current state, $s_{l,t_c}$. To facilitate the application of deep reinforcement learning techniques, we represent each coalition structure $s_l$ as an $N \times N$ binary adjacency matrix, flattened into a vector where $(i, j) = 1$ if agents $i$ and $j$ are in the same coalition. This compact representation remains equivariant under consistent permutations of agent indices. Actions are merge decisions of arbitrary two coalitions or a no-merge option, encoded as an $(N + 1)$-dimensional two-hot vector. The encoding dimension thus scales linearly with $N$, avoiding quadratic growth in network outputs while uniquely representing each merge, which helps maintain scalability and stable learning. An illustration is provided in the Appendix D.1.

This matrix-based state representation facilitates generalization to coalition formation problems with larger agent populations by encoding coalition structures into a vector that systematically captures all pairwise relationships among agents. Such representation provides a consistent input format for deep learning architectures. Crucially, this input format is equivariant under consistent permutations of agent indices, and by fixing the agent ordering during training and evaluation we ensure stable performance. The framework enables the model to extract generalizable patterns regarding optimal agent grouping strategies. For instance, by acquiring these fundamental coalition formation principles from smaller problem instances, the neural architecture develops transferable knowledge that can be systematically applied to navigate the combinatorially complex space of possible coalition structures when scaling to larger agent populations (e.g., the learned $\{\{1, 2\}, \{3\}\}$ coalition structure value could be transferred to $\{\{1, 2\}, \{3\}, \{4, 5\}\}$).

### 4.2 BI-LEVEL MARKOV GAME FOR CSG

The traditional CSG problem focuses on identifying a coalition structure that maximizes social welfare. This concept is also highly relevant to multi-agent reinforcement learning tasks, where coalition formation facilitates efficient coordination by minimizing redundant efforts in applications such as warehouse management. However, in practical scenarios, the value of a coalition structure is often influenced by learning agents. For example, agents must adapt and optimize their policies in response to a given coalition structure, making the problem inherently dynamic. This requires the algorithm to form the coalition structure dynamically. This interdependence between coalition formation and policy optimization can be naturally framed as a bi-level optimization problem. Our framework, however, adopts the bi-level optimization concept by redefining the leader's objective specifically to maximize the value of the coalition structures. This shift in focus aligns with the hierarchical nature of the problem, where the leader prioritizes the global optimization of coalition formation, while the followers engage in local cooperative optimization to realize their individual and collective goals. In this context, the leader is responsible for dynamically determining the optimal partition of follower agents into coalitions to maximize the overall value of the coalition structure. Meanwhile, the agents $i \in N$, acting as followers, aim to optimize the value of their respective coalitions within the constraints established by the leader's partitioning.

By balancing these two levels of optimization, our framework provides an efficient framework for solving the CSG problem. The followers phase can be modeled as a **cooperative game**, wherein agents work together to maximize the value of their respective coalitions. In addition, the leader phase can be formulated as an episodic MDP, reflecting the sequential decision-making with the goal of generating coalition structures dynamically. Therefore, the optimization objectives for the leader and followers in the Bi-level MDP can be defined as:

**Definition 4.1.** In the bi-level optimization problem, the leader and followers aim to solve the following bi-level optimization problem respectively: Given each state $s_l$, define the set of Nash equilibrium for set of player $C \in s_l$:

$$\text{NE}(s_l) = \{\pi_f \mid J_f^C(\pi_f^C, \pi_f^{-C}) \geq J_f^C(\pi_f'^C, \pi_f^{-C}), \forall C \in s_l\}.$$

Where $\pi_f^{-C}$ represents the policy of all agents not in $C$. Given the set of Nash equilibrium of followers' policy, define the objective of leader policy as follows:

$$\max_{\pi_l \in \Pi_L} J_l(\pi_l, \pi_f) \quad \text{s.t. } \pi_f = \arg\max_{\pi_f \in \text{NE}(s_l)} \sum_{C \in s_l} J_f^C(\pi_f).$$

The above constraints on $\pi_f$ imply that, given the current coalition structure $s_l$, each coalition $C \in s_l$ acts as a single rational agent. We further assume that the players corresponding to coalitions in $s_l$ will choose the optimal Nash equilibrium if multiple Nash equilibria exist. We adopt a measurable selection rule $\sigma : \text{NE}(s_l) \to \Pi_f$ that selects the equilibrium yielding the highest coalition structure reward for each $s_l$, which ensures a deterministic upper-level transition reward mapping. In practice, MADDPG-trained followers provide approximate best responses; Figure 4 demonstrates robustness to moderate approximation errors.

**Bi-level Optimization** In this bi-level optimization framework (Figure 1), the leader and the followers operate at the upper level and the lower level, respectively. The tree structure is adapted from (Sandholm et al., 1999). The leader aims to identify the best coalition structure by learning the optimal policy $\pi_l \in \Pi_L$, which maximizes the value of the coalition structure. The upper-level optimization incorporates the best responses of the followers, represented by their value functions $V_f^{i,*}$. The leader's objective function $J_l(\pi_l, \pi_f)$ depends on the leader's action $a_l$, the current coalition structure $s_l$, and the followers' policies $\pi_f^i$. The constraint ensures that the leader's optimization accounts for the followers' best response $\pi_f^i$, which maximize their respective objectives. At the lower level, each follower agent $i \in C, C \in CS$ independently optimizes its own value function $J_f^i$ based on its coalition value $r_{f,t}^C$. The goal of each follower is to identify the optimal action $a_f^i \in A_f$ in the sense of coalition value, given the coalition structure defined by the leader's state $s_l$ and its current state $s_f$.

Based on the definition of the reward function (Equation 1), we can further define the action value functions for the leader in the CSG process. In our framework, the leader agent orchestrates the CSG in high level, while the follower agents operate in the lower-level optimization process, aiming to maximize their coalition rewards. Let $Q_f$ and $Q_l$ denote the action-value functions for the followers and the leader, respectively. Then, the optimal action value function for the leader is defined as:

**Definition 4.2.** The optimal action value function for the leader agent is:

$$Q_l^*(s_l, a_l, \pi_f) := r_l(s_l, a_l, \pi_f) + \max_{a_l'} Q_l^*(\mathcal{T}_l(s_l, a_l), a_l', \pi_f).$$

Based on the definition of the action value function and Bellman operator of the leader agent, we could further define the followers' optimal action value function as follows:

**Definition 4.3.** The optimal action value function for the follower agents is defined as:

$$Q_f^{i,*}(s_f, a_f, s_l) := r_f^C(s_f, a_f, s_l) + \max_{a_f'} Q_f^{i,*}(\mathcal{T}_f(s_f, a_f), a_f', s_l).$$

Due to space constraints, we defer the detailed theoretical analysis in Appendix F, where we focus on the planning problem under the MDP formulation of CSG and establish the correctness of our RL-based approach. In particular, we establish the correctness of our RL-based approach by proving that it converges to optimal coalition structures under standard assumptions. This analysis offers a rigorous foundation for applying reinforcement learning to coalition structure generation problems.

**Algorithm Overview** In Algorithm 1, we introduce **B**i-level **R**einforcement Learn**I**ng for **D**ynamic **G**roup Structur**E** (BRIDGE) algorithm, a hierarchical framework for optimizing both coalition structures and agent strategies in multi-agent systems. Following the meta-reinforcement learning

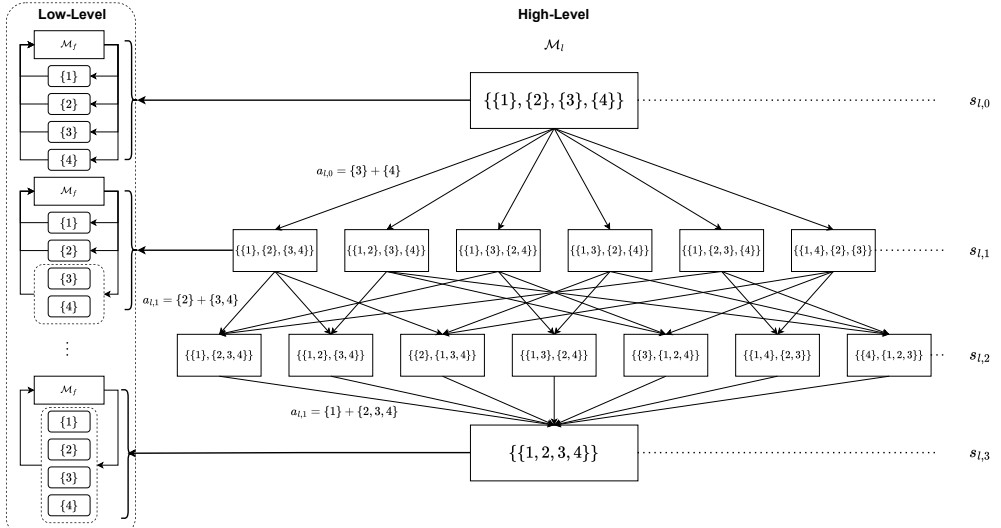

Figure 1: Bi-level optimization process for the leader agent and follower agents. In the high-level optimization process, the leader agent determines the coalition structure of the follower agents and it always starts from singleton coalition ($s_{l,0} = \{\{1\}, \{2\}, \{3\}, \{4\}\}$). Then the leader will perform the action $a_{l,0}$ which leads to the next leader state. The reward is calculated by the difference between the value of the coalition structures. This value is achieved by the optimization process in the low-level, where each follower agent optimizes its coalition reward. The high-level episode lasts for at most $N-1$ merge decisions. Because a null action is permitted at any step, the process may end with a non–grand coalition partition.

paradigm, the algorithm's nested optimization structure consists of two levels: the upper level serves as a meta-policy that learns to propose effective coalition structures, while the lower level learns the optimal strategies within the proposed structure. By employing neural networks at the meta-level, our method efficiently adapts the coalition structure based on the performance of lower-level policies, enabling efficient searching across different group configurations. The ultimate objective is to determine the optimal coalition structure and the optimal coalition strategies for all participating agents.

The algorithmic process (Algorithm 1) is structured around two nested iterative procedures. The outer loop is indexed by iteration counter $c$. In each step $t_c$, the leader agent produces a coalition structure transition tuple $\{(s_{l,t_c}, a_{l,t_c}, s_{l,t_c+1}, r_{l,t_c})\}$ using the leader agent's Q-function $Q_l$. This outer loop provides the framework for optimization for the coalition structures. Within each outer loop iteration, the inner loop (indexed by episode counter $t_k$) focuses on optimizing follower agents behaviors within the given coalition structure. The follower agents' optimization process adapts the MADDPG framework (Lowe et al., 2017) by integrating stochastic policies in place of the original deterministic ones. During each episode, the followers' actors generate transition tuples $\{(\boldsymbol{o}^C_{f,t_k}, \boldsymbol{a}^C_{f,t_k}, \boldsymbol{o}^C_{f,t_k+1}, \boldsymbol{r}^C_{f,t_k})\}$. These transitions are then used to update the critic $Q^i_f$, which evaluates the quality of actions. Subsequently, the actor is updated using the policy gradient, which is informed by the Critic's evaluations, to improve the agents' policies. Following the completion of each inner loop, the algorithm calculates the leader's Q network loss $L(\theta_l)$. The leader's Q function parameters are subsequently updated using the second learning rate $\rho_l$. This bi-level approach enables the joint optimization of both coalition structures and individual agent strategies. The slower outer loop focuses on coalition formation while the faster inner loop optimizes agent behaviors within established coalitions. This hierarchical learning structure allows the algorithm to effectively address the complex interdependencies between coalition formation and agent strategy optimization in multi-agent systems.

## 5 NUMERICAL EVALUATION

In the experimental evaluation, we aim to demonstrate three key advantages of the BRIDGE framework when applied to CSG problems:

---

**Algorithm 1** BRIDGE

---

**Input**: followers' replay buffer $\mathcal{B}$, leader's replay buffer $\mathcal{G}$, learning rate $\rho_{f,\text{actor}}, \rho_{f,\text{critic}}, \rho_l$,

**Output**: approximate coalition structure $CS^*$, approximate coalition strategies for each agent $\boldsymbol{\pi}_f$

1: Initialize leader Q-network $Q_l(s_l, a_l; \theta_l)$.
2: Initialize follower actor $\pi_{\theta_f^i}$ and critic $Q_f^i$ networks for each follower $i$ with random weights.
3: **for** Iteration $c = 0, 1, 2 \ldots, C$ **do**
4:     **for** Coalition time step $t_c = 0, 1, 2 \ldots N - 1$ **do**
5:         Select a random leader action $a_{l,t_c}$ with probability $\epsilon$, otherwise select $a_{l,t_c} = \arg\max_{a_{l,t_c}} Q_l(s_{l,t_c}, a_{l,t_c}; \theta_l)$.
6:         Observe next state $s_{l,t_c+1}$ and reward $r_{l,t_c}$.
7:         Push transitions $\{(s_{l,t_c}, a_{l,t_c}, s_{l,t_c+1}, r_{l,t_c})\}$ into replay buffer $\mathcal{G}$.
8:         **for** Episode $t_k = 0, 1, 2 \ldots K$ **do**
9:             For agent $i = 1 : N$, play $\boldsymbol{a}_{f,t_k}^{CS_c} \sim \boldsymbol{\pi}_f$ based on current coalition structure $CS_c$.
10:           Observe new observation $\boldsymbol{o}_{f,t_k+1}^{CS_c}$, reward $\boldsymbol{r}_{f,t_k}^{CS_c}$.
11:           Push transitions $\{(\boldsymbol{o}_{f,t_k}^{CS_c}, \boldsymbol{a}_{f,t_k}^{CS_c}, \boldsymbol{o}_{f,t_k+1}^{CS_c}, \boldsymbol{r}_{f,t_k}^{CS_c})\}$ into replay buffer $\mathcal{B}$.
12:           Sample a minibatch of $|\mathcal{B}_{\text{mini}}|$ transitions $\{(\boldsymbol{o}_{f,b}^{CS_c}, \boldsymbol{a}_{f,b}^{CS_c}, \boldsymbol{o}_{f,b}'^{,CS_c}, \boldsymbol{r}_{f,b}^{CS_c})\}$ from $\mathcal{B}$.
13:           Compute the target actions from the target policy: $\boldsymbol{a}_{f,b}' \sim \boldsymbol{\pi}_f(\cdot|\boldsymbol{o}_{f,b}')$.
14:           Compute the target Q-values: $y_b^i = \boldsymbol{r}_{f,b}^C + \gamma Q_f^i(\boldsymbol{o}_{f,b}'^{,C}, \boldsymbol{a}_{f,b}'^{,C}; \theta_f^i)$, for $i \in C, C \in CS_c$.
15:           Compute the critic loss: $L(\theta_f^i) = \frac{1}{|\mathcal{B}_{\text{mini}}|} \sum_b \left(y_b^i - Q_f^i(\boldsymbol{o}_{f,b}^C, \boldsymbol{a}_{f,b}^C; \theta_f^i)\right)^2$.
16:           Update the critic by minimizing the loss: $\theta_f^i \leftarrow \theta_f^i - \rho_{f,\text{critic}} \nabla_{\theta_f^i} L(\theta_f^i)$.
17:           Calculate the policy gradient: $\nabla_{\theta_f^i} J(\theta_f^i) = \mathbb{E}_{\boldsymbol{a}_{f,b} \sim \pi_f} \left[\nabla_{\theta_f^i} \log \pi_{\theta_f^i}(a_{f,b}^i|o_{f,b}^i) Q_f^i(\boldsymbol{o}_{f,b}^C, \boldsymbol{a}_{f,b}^C; \theta_f^i)\right]$.
18:           Update the actor using the policy gradient: $\theta_f^i \leftarrow \theta_f^i + \rho_{f,\text{actor}} \nabla_{\theta_f^i} J(\theta_f^i)$.
19:         **end for**
20:     **end for**
21:     Sample a minibatch of $|\mathcal{G}_{\text{mini}}|$ transitions $\{(s_{l,g}, a_{l,g}, s_{l,g}', r_{l,g})\}$ from $\mathcal{G}$.
22:     Compute the target Q-values: $y_g = r_{l,g} + \gamma \max_{a'} Q_l(s_{l,g}', a'; \theta_l)$.
23:     Compute the Q-network loss: $L(\theta_l) = \frac{1}{|\mathcal{G}_{\text{mini}}|} \sum_g (y_g - Q_l(s_{l,g}, a_{l,g}; \theta_l))^2$.
24:     Update the leader Q-network parameters by minimizing the loss: $\theta_l \leftarrow \theta_l - \rho_l \nabla_{\theta_l} L(\theta_l)$.
25: **end for**

---

- **Enhanced Generalizability:** We investigate the capacity of BRIDGE to effectively generalize learned strategies from problem instances with a smaller number of agents to those involving larger agent population(Table 1). We also provide a comparative analysis on our method and other baselines as well(Table 3).

- **Efficient Inference:** We assess the computational efficiency of BRIDGE by comparing its inference time to that of established heuristic CSG methods, highlighting its potential for rapid solution generation(Table 2).

- **Performance improvement in mixed-motive Games:** We evaluate the performance of BRIDGE to address and solve CSG problems characterized by both normal-form(Figure 2 and Figure 7) and Markov games(Figure 3 and Figure 4).

We compared our algorithm with C-Link (Farinelli et al., 2013), GRASP (Di Mauro et al., 2010), CSG-UCT (Wu & Ramchurn, 2020), SALDAE (Taguelmimt et al., 2024), which are currently the leading approximate methods for the CSG problem. We provide further ablation studies with respect to our neural network size in Appendix D.3.3.

## 5.1 COMMON CSG BENCHMARK PROBLEMS

A standard approach to evaluating CSG algorithms involves selecting representative problem instances and comparing various algorithms without providing them with prior knowledge of the types of utilities they will encounter. While our algorithm is capable of solving CSG problems

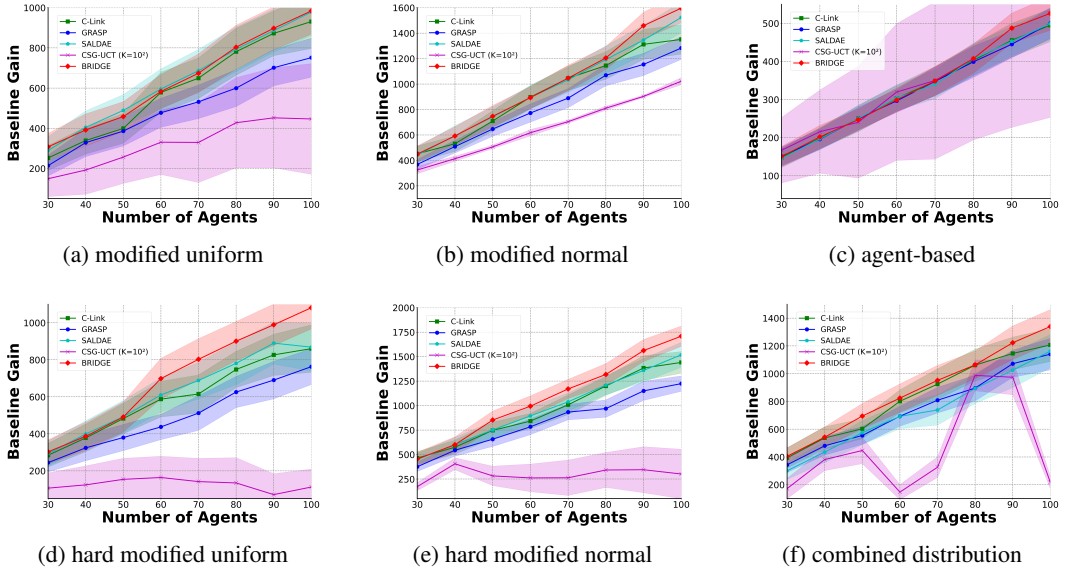

Figure 2: Coalition structure value comparison across six different value distributions: (a) modified uniform, (b) modified normal, (c) agent-based, (d) hard modified uniform, (e) hard modified normal, and (f) combined distribution scenarios. The experiments compare four coalition formation baselines with varying numbers of agents (30-100), comparing their achieved coalition structure values with our algorithm. For each baseline, 10 independent experimental runs were conducted to compute the mean and standard deviation of the resulting coalition structure values,

across different utility types, we benchmark its performance using three commonly studied utility distributions and three harder utility distributions. The three traditional CSG problems are defined as: **Modified Uniform**(Adams et al., 2010): $v(C) \sim U(0, 10 \times |C|)$, and $v(C)$ is increased by a random number $r \sim U(0, 50)$ with probability 0.2. **Modified Normal**(Rahwan et al., 2012): $v(C) \sim N(10 \times |C|, 0.1^2)$, and the coalition value $v(C)$ is increased by a random number $r \sim U(0, 50)$ with probability 0.2. **Agent-based**(Rahwan et al., 2012): $v(C) = \sum_{i \in C} p_C^i$ where $p_C^i \sim U(0, 2p_i)$ and $p_i \sim U(0, 10)$ is a random power for agent $i$. We also introduce three additional, more challenging utility distributions(hard modified uniform, hard modified normal, combined distribution) that incorporate increased stochasticity and implement penalties for larger coalition sizes, thereby providing a more rigorous evaluation framework that better approximates real-world coalition formation scenarios. Detailed specifications of the value functions of these scenarios are presented in Appendix D.2.

To evaluate the generalization capability of our algorithm, a primary advantage posited in this work, we present the performance results across varying agent populations instances(5 to 10) under modified normal distribution in Table 1. Following an initialization of the network architecture to accommodate the 10-agent scenario, the algorithm was initially trained on instances involving 3 agents and subsequently evaluated in environments with 5 to 10 agents. Optimal coalition structure values for these scenarios were determined via brute-force search, and the table reports the percentage of the achieved value relative to these optima. While the few-shot conditions involve continued training on the target agent instances for 100 and 200 episodes, respectively. As evidenced in Table 1, our method, demonstrates a substantial improvement in performance compared to a random policy across all evaluated agent numbers. The few-shot training paradigms yield progressively higher performance, with the 200-episode fine-tuning achieving the highest value percentages. These results underscore the robust generalizability of BRIDGE to coalition structure generation problems with a larger number of agents than those encountered during initial training. We also provide a comparative analysis on other traditional coalition structure baselines performance with the same budget in Appendix. BRIDGE trains for 300 outer-loop episodes with early stopping when reward variance stays below 1% for 20 episodes.

Table 1: Evaluation of the generalization performance of different training paradigms on coalition structure generation tasks(modified normal distribution) across varying numbers of agents (5 to 10), with models pre-trained on instances with 3 agents.

| Pretrain on 3 agents | 5 agents | 6 agents | 7 agents | 8 agents | 9 agents | 10 agents |
|---|---|---|---|---|---|---|
| Random Policy | 44.42% | 36.59% | 36.32% | 30.97% | 25.05% | 21.01% |
| Few-Shot (100 episode) | 88.83% | 66.17% | 59.67% | 55.62% | 47.18% | 41.61% |
| Few-Shot (200 episode) | **97.2%** | **80.33%** | **78.68%** | **63.52%** | **58.15%** | **48.24%** |

Figures 2a - 2f summarize our experimental results on the common benchmark problems with utilities using the distributions described above. As seen in the figures, our method outperforms other algorithms with better solution quality in all tested instances. We use the Baseline Gain metric (Baseline Gain $= v(CS) - \sum_{i \in \mathcal{N}} v(\{i\})$) to quantify improvement over singleton coalitions in complex environments where optimal solutions are intractable. In particular, in hard modified uniform and hard modified normal experiment, our method can find better results than any other baseline algorithms. This set of experiments demonstrates the advantage of our method compared to state-of-the-art approximate methods. In order to further study how the leader agent's ability evolve over training, we propose visualization results in Appendix D.3 Figure 7.

## 5.2 MIXED-MOTIVE MARKOV GAMES

In the LBF (Level-Based Formation) environment (Figure 3), tasks are represented by apples with varying sizes that correspond to different resource requirements, where apples randomly spawn across the map. The leader agent must strategically select follower agents to complete tasks while maximizing rewards and each follower agent recruitment incurs a cost. Our empirical evaluation compares

| LBF | Singleton | C-Link | GRASP | SALDAE | CSG-UCT | BRIDGE |
|---|---|---|---|---|---|---|
| 6a4t | 28.20±1.5 | 27.70±1.2 | 9.17±0.8 | 25.00±1.3 | 31.00±1.6 | **34.17±1.4** |
| 6a5t | 31.01±1.4 | 32.08±1.1 | 11.90±0.7 | 27.50±1.5 | **34.75±1.8** | 33.11±1.2 |
| 8a4t | 32.95±1.7 | 32.58±1.4 | 12.44±0.9 | 27.43±1.3 | 34.76±1.5 | **38.57±1.6** |
| 8a5t | 43.18±1.8 | 40.68±1.5 | 23.33±1.0 | 31.33±1.6 | 45.33±1.9 | **52.17±2.1** |
| 10a4t | 70.62±2.0 | 66.23±1.8 | 29.71±1.2 | 41.33±1.7 | 73.33±2.3 | **77.43±2.4** |
| 10a5t | 75.23±2.2 | 73.71±1.9 | 33.10±1.3 | 63.83±2.0 | 82.83±2.5 | **89.29±2.6** |

Figure 3: Mixed-motive Markov game (LBF) across six different scenarios. Values are reported as mean $\pm$ standard deviation over multiple runs. 'a' represents agents, 't' represents tasks (apples). The value is defined as the baseline gain.

our framework against traditional CSG baselines across multiple configurations. Since traditional CSG baselines are not originally designed for the mixed-motive Markov games. We adopt the same training paradigm as BRIDGE by calculating the coalition structures value using low-level agents. In the Singleton baseline, each agent operates independently as a self-interested decision maker. Overall, this ablation verifies that although our theory assumes an optimal follower for clarity of exposition, the practical system is robust to moderate follower suboptimality and only fails when the follower is so under-trained that it no longer provides reliable local value signals.

To assess our leader agent's robustness to follower policy quality, we conducted an ablation study by pairing a leader with follower policies of varying training epochs. The findings demonstrate that the leader's performance remains high and stable with well-converged followers (10-15 epochs) and degrades accordingly with moderately trained ones (7 epochs), showcasing resilience to approximation errors. As expected, performance drops sharply when paired with

| LBF | 15 epochs | 10 epochs | 7 epochs | 5 epochs |
|---|---|---|---|---|
| 6a4t | 32.11 | 34.17 | 26.73 | 2.82 |
| 6a5t | 36.87 | 33.11 | 34.51 | 2.47 |
| 8a4t | 37.05 | 38.57 | 21.46 | 9.84 |
| 8a5t | 51.24 | 52.17 | 47.44 | 11.49 |
| 10a4t | 77.54 | 77.43 | 69.78 | 5.38 |
| 10a5t | 91.28 | 89.29 | 88.35 | 29.97 |

Figure 4: Comparative performance of the leader to follower approximation errors.

poorly trained followers (5 epochs) that provide unreliable reward signals. This study confirms that while our framework theoretically assumes follower optimality, it is practically robust to moderate errors.

## 6 CONCLUSION

This paper introduces BRIDGE, a bi-level reinforcement learning framework that jointly optimizes coalition structures and agent policies. The framework dynamically adapts to changing conditions, and its theoretical formulation aligns the RL objective with optimal coalition formation. Empirically,

BRIDGE performs well across both classic CSG problems and mixed-motive Markov games. These results highlight the promise of bi-level reinforcement learning for multi-agent optimization tasks involving hierarchical decision-making, opening new opportunities for research in coalition formation and multi-agent systems.

## ACKNOWLEDGEMENT

This work was supported by the Engineering and Physical Sciences Research Council [grant number EP/Y003187/1 and UKRI849].

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

## A    DECLARATION OF THE USE OF LARGE LANGUAGE MODELS (LLMS)

We utilized large language model (LLM) to assist in proofreading and improving the language, grammar, and clarity of this manuscript. The authors retain full responsibility for all intellectual content and claims presented.

## B    ETHICS STATEMENT

## C    NOTATION TABLE

We provide the notation tableC here for the readers to check on the physical meaning of the notations.

| Symbol | Definition |
|---|---|
| *Coalition Structure Generation* | |
| $N$ | Number of agents |
| $\mathcal{N} = \{1, \ldots, N\}$ | Set of agents |
| $C \subseteq \mathcal{N}$ | Non-empty coalition (subset of agents) |
| $CS = \{C_1, \ldots, C_k\}$ | Coalition structure (partition of $\mathcal{N}$) |
| $\Pi(\mathcal{N})$ | Set of all possible coalition structures |
| $v : 2^{\mathcal{N}} \to \mathbb{R}$ | Characteristic function mapping coalitions to values |
| $v(CS) = \sum_{C \in CS} v(C)$ | Value of coalition structure |
| *Leader (Upper-Level) MDP* | |
| $\mathcal{M}_l = \langle S_l, T_l, A_l, r_l, H_l \rangle$ | Episodic MDP for coalition formation |
| $S_l = \Pi(\mathcal{N})$ | State space (coalition structures) |
| $s_{l,0} = \{\{1\}, \ldots, \{N\}\}$ | Initial state (singleton partition) |
| $a_l = (C_i, C_j)$ or $\varnothing$ | Action: merge two coalitions or stop |
| $T_l : S_l \times A_l \to S_l$ | Deterministic merge transition |
| $r_l(s_l, a_l, \pi_f)$ | Reward: coalition value difference (Eq. 1) |
| $H_l = N - 1$ | Maximum number of merge steps |
| $\pi_l, \pi_l^*$ | Leader policy and optimal policy |
| $Q_l(s_l, a_l; \theta_l)$ | Leader action-value function |
| $J_l(\pi_l, \pi_f)$ | Expected cumulative reward |
| *Follower (Lower-Level) Game* | |
| $\mathcal{M}_f = \langle \mathcal{N}, S_f, T_f, A_f, r_f, \gamma_f \rangle$ | Cooperative Markov game among followers |
| $s_f \in S_f$ | Environment state |
| $\mathbf{a}_f = [a_f^1, \ldots, a_f^N]$ | Joint action of all followers |
| $T_f(s' \mid s, \mathbf{a}_f)$ | Environment transition dynamics |
| $r_f^C(s_f, \mathbf{a}_f)$ | Instantaneous reward of coalition $C$ |
| $\gamma_f \in (0, 1)$ | Discount factor |
| $\pi_f, \pi_f^*$ | Follower joint policy and optimal policy |
| $J_f^C(\pi_f)$ | Expected discounted return of coalition $C$ |
| $Q_f^i(s_f, a_f; \theta_f^i)$ | Follower $i$'s action-value function |

# D  DETAILS ON EXPERIMENT SETTINGS

## D.1  ILLUSTRATION FOR Q-NETWORK AND TRANSITION PROCESS

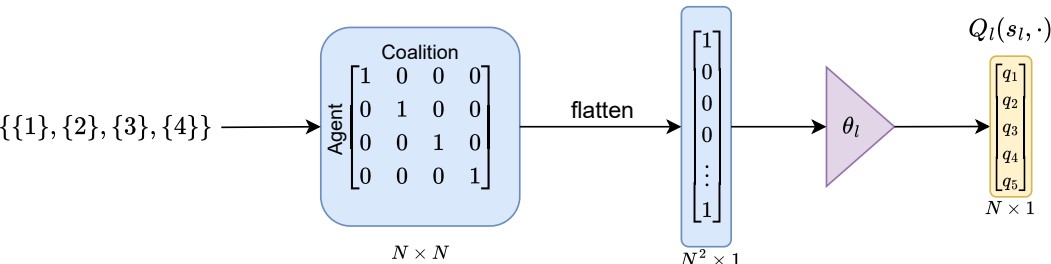

Figure 5: Illustration of the architecture of the Q-network used by the leader agent.

Figure 5 illustrates the Q-network architecture employed by the leader agent for making coalition formation decisions. The process begins with the initial coalition structure state, such as $\{\{1\}, \{2\}, \{3\}, \{4\}\}$. This state is transformed into an $N \times N$ binary adjacency matrix, where an entry $(i, j)$ of 1 denotes that agents $i$ and $j$ belong to the same coalition. This matrix is then flattened into an $N^2 \times 1$ vector, creating a structured input that captures all coalition relationships while remaining equivariant to consistent permutations of agent indices. The vector feeds into the

Q-network (parameterized by $\theta_l$), which processes this information to evaluate potential merge actions. The network's output, visualized as the yellow vector, is a fixed-size $(N+1) \times 1$ array. This vector represents latent values used to compute the Q-values for all valid actions, rather than directly representing the Q-values themselves. Let $z(s) \in \mathbb{R}^{N+1}$ denote the output of the leader Q-network at state $s$. For any legal merge $a = (C_i, C_j)$ with $i < j$, we compute action values by composition

$$Q_l(s, a) = \psi\big(z_i(s), z_j(s)\big), \qquad Q_l\big(s, \varnothing\big) = z_{N+1}(s),$$

where $\psi$ is a shared pairwise scorer (e.g., additive). We enforce legality via masking and operate only over the legal set in both selection and backup:

$$a^\star = \arg \max_{a \in \mathcal{A}(s)} Q_l(s, a), \qquad y = r + \gamma \max_{a' \in \mathcal{A}(s')} \overline{Q}_l(s', a').$$

The leader agent selects the pair $(i, j)$ that maximizes this computed Q-value, after masking any illegal or unused indices. The final element of the output vector represents the Q-value for the null action (to not merge). This architecture enables the leader agent to make informed decisions through an $O(N)$-dimensional encoding of actions, which avoids quadratic growth in network outputs even though the number of feasible merge pairs is $O(N^2)$. This design keeps the framework scalable while effectively guiding the coalition formation process toward arrangements that maximize collective utility.

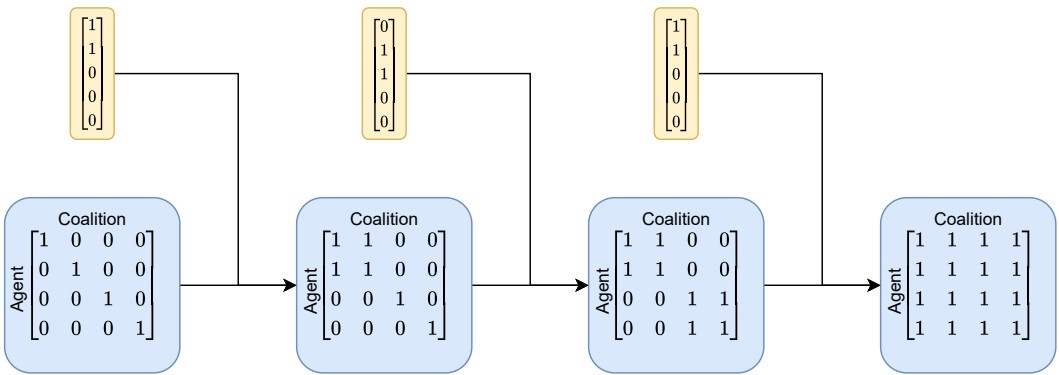

Figure 6: Illustration of the architecture of the transition of the leader agent.

Figure 6 illustrates the transition process employed by the leader agent in dynamic coalition formation, depicting the generated process from an initial coalition structure towards the final coalition structure. Beginning with the fully singleton coalition $\{\{1\}, \{2\}, \{3\}, \{4\}\}$ at the leftmost position, the system employs dual representational components: blue adjacency matrices that encode coalition membership relationships (with 1s indicating agents in the same coalition), and corresponding yellow action vectors $a_i$ that guide action selection. Notably, in the second state after the first merger ($\{\{1, 2\}, \{3\}, \{4\}\}$), the system contains only three coalitions, making the fourth dimension of action vector $a_1$ illegal and therefore unavailable for selection—this constraint guides the agent toward valid coalitional structures only. The sequence proceeds with subsequent mergers joining agent 3 with the $\{1, 2\}$ coalition to form $\{\{1, 2, 3\}, \{4\}\}$ and finally incorporating agent 4 to achieve grand coalition in $\{\{1, 2, 3, 4\}\}$, where the adjacency matrix contains all 1s indicating universal coalition membership. Throughout this evolution, the architecture demonstrates how reinforcement learning enables the leader agent to navigate complex coalition spaces within operational constraints, making sequential decisions that optimize toward the system's collective objectives.

## D.2 DETAILS ON CSG PROBLEMS

Below we detail the coalition value functions used in Section 5.1. Throughout, let $C \subseteq \{1, \ldots, N\}$ denote a coalition and $k := |C|$ its size. We write $\mathcal{U}(a, b)$ for the uniform distribution on $[a, b]$. For normal draws, we use the "scale" parameterization $\mathcal{N}(\mu, \varsigma)$ where the second argument $\varsigma > 0$ denotes the scale (not variance). Global parameters are: baseline factor $\alpha > 0$, boost probability $p \in [0, 1]$, interaction factor $\gamma \geq 0$, and size penalty $\delta \geq 0$. The history factor is $h \sim \mathcal{U}(0.8, 1.2)$. For compactness, we define

$$p_{\text{int}}(k) := \gamma\, k^2, \quad p_{\text{size}}(k) := \delta\, (k - 4)^2\, \mathbf{1}[k > 4], \quad p_{\text{coal}}(k) \sim \mathcal{U}(0, 2k)\, \mathbf{1}[k > 6].$$

**Hard Modified Uniform Distribution.** Draw a base value $b \sim \mathcal{U}(0, \alpha k)$. With probability $p$, draw a boost $r_b^{\text{uni}} \sim \mathcal{U}(0, 40)$ (otherwise $r_b^{\text{uni}} = 0$). The coalition value is

$$V(C) = \max\Big(0, \ h\left[b + r_b^{\text{uni}} - p_{\text{int}}(k) - p_{\text{size}}(k) - p_{\text{coal}}(k)\right]\Big).$$

**Hard Modified Normal Distribution.** Draw a base value $b \sim \mathcal{N}(0, 2k)$ (scale $2k$). With probability $p$, draw a boost $r_b^{\text{norm}} \sim \mathcal{N}(0, 40)$ (scale 40; otherwise $r_b^{\text{norm}} = 0$). The coalition value is

$$V(C) = \max\Big(0, \ h\left[b + r_b^{\text{norm}} - p_{\text{int}}(k) - p_{\text{size}}(k) - p_{\text{coal}}(k)\right]\Big).$$

**Combined Distribution.** Draw two base values $b_{\text{uni}} \sim \mathcal{U}(0, \alpha k)$ and $b_{\text{norm}} \sim \mathcal{N}(0, 2k)$. With probability $p$, add independent boosts $r_b^{\text{uni}} \sim \mathcal{U}(0, 40)$ and $r_b^{\text{norm}} \sim \mathcal{N}(0, 40)$ (otherwise each is 0). The coalition value is

$$V(C) = \max\Big(0, \ h\left[b_{\text{uni}} + b_{\text{norm}} + r_b^{\text{uni}} + r_b^{\text{norm}} - p_{\text{int}}(k) - p_{\text{size}}(k) - p_{\text{coal}}(k)\right]\Big).$$

### D.3 ADDITIONAL EXPERIMENT RESULTS

In order to further evaluate the learning dynamics of the BRIDGE framework, we provide additional results on evaluating the optimal gain across outer iterations. The optimal gain($\max v(CS)$) is defined as the best coalition structure value algorithm could get under the present agent number. The results demonstrate the achievable optimal gain scales positively with the number of agents, as expected in a system with increasing resources. This learning process appears to converge around 300 steps; the mean gain plot, box plot distributions, and heatmap all show that performance gains become marginal after this point, with the results for steps 300, 400, and 500 being nearly indistinguishable. While the average performance stabilizes, the standard deviation plots and the widening distributions in the box plot reveal that the variability of the optimal gain increases in larger systems, indicating a more complex and diverse solution landscape as the problem scales.

#### D.3.1 INFERENCE SPEED

The efficiency of inference following the training phase is another notable advantage of our framework. Table 2 presents a comparative analysis of the inference times, measured in seconds, for BRIDGE and several baseline coalition structure generation algorithms under the modified normal distribution setting. While our method's inference speed is comparable to or slightly slower than other heuristic methods for lower agent counts (30-50), it exhibits significantly improved performance as the number of agents increases (60-100). Notably, the inference time of our method demonstrates a considerably less pronounced scaling trend with the number of agents compared to the heuristic baselines, particularly CSG-UCT. This characteristic suggests a greater potential for BRIDGE to be effectively applied to coalition structure generation problems involving larger agent populations.

Table 2: Comparative analysis of inference times (in seconds) for coalition structure generation(modified normal distribution) using baseline algorithms and the proposed BRIDGE framework across varying numbers of agents (30 to 100).

| Time(s) | 30 agents | 40 agents | 50 agents | 60 agents | 70 agents | 80 agents | 90 agents | 100 agents |
|---|---|---|---|---|---|---|---|---|
| C-Link | 3.44 | 4.49 | 7.11 | 10.60 | 14.28 | 18.77 | 24.42 | 30.44 |
| GRASP | **0.37** | **0.88** | 2.93 | 5.81 | 8.19 | 10.95 | 19.86 | 26.47 |
| SALDAE | 0.56 | 1.57 | **2.54** | 5.30 | 8.61 | 14.13 | 19.74 | 28.31 |
| CSG-UCT | 2.24 | 7.10 | 17.19 | 35.90 | 68.15 | 116.18 | 194.01 | 303.79 |
| BRIDGE | 1.09 | 1.98 | 2.83 | **3.62** | **4.90** | **5.67** | **6.84** | **8.72** |

#### D.3.2 ADDITIONAL GENERALIZATION ABILITY ANALYSIS

Traditional Coalition Structure Generation (CSG) baselines, such as C-Link, GRASP, and CSG-UCT, are typically search-based or heuristic algorithms. Their methodologies are inherently designed for a fixed number of agents, requiring the problem size to be predefined to construct the search space. Consequently, these methods do not possess an intrinsic mechanism to be trained on a 3-agent

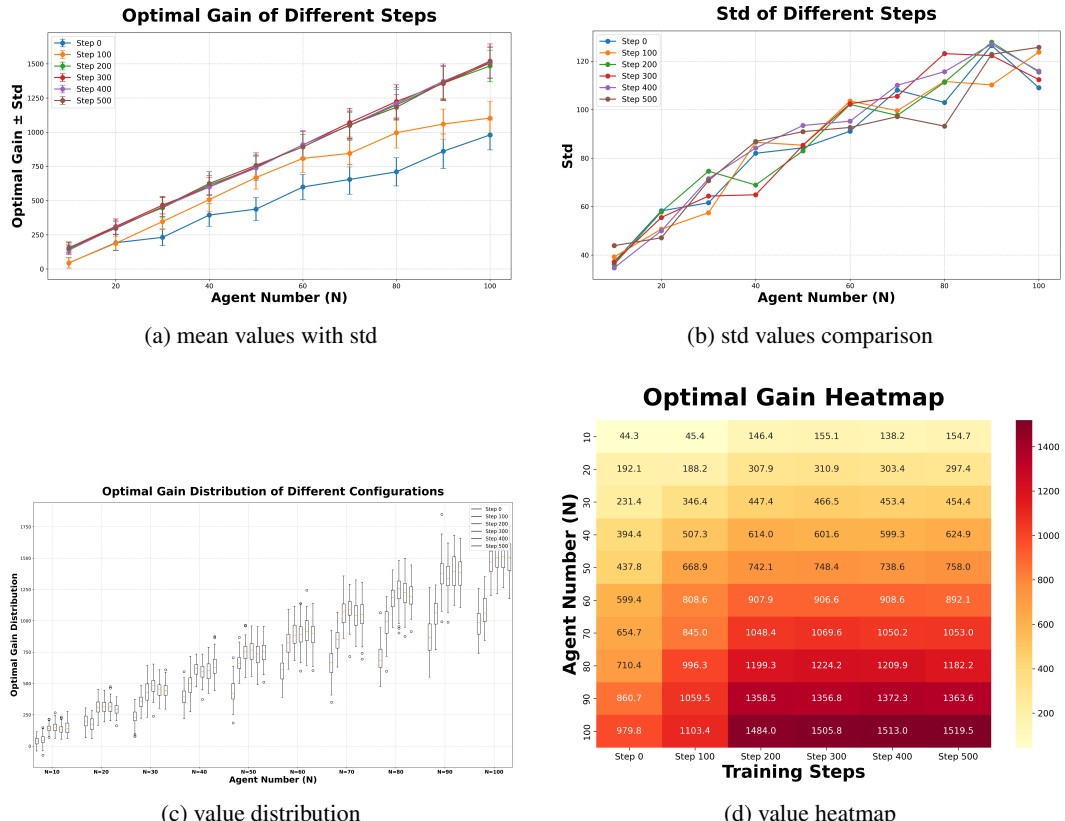

Figure 7: This set of visualizations illustrates the variation in optimal gain across different training steps and agent numbers. Subfigure (a) shows the mean optimal gain with standard deviation, indicating a gradual increase in gain as the number of agents grows. Subfigure (b) compares the standard deviation across different training steps, revealing fluctuations in multi-agent systems of varying sizes. Subfigure (c) presents a box plot of optimal gain distributions, showing an increasing spread as the agent number rises. Finally, subfigure (d) provides a heatmap representation of optimal gain evolution, where darker colors indicate higher gains. Together, these visualizations highlight the impact of training steps and agent numbers on optimal gain.

instance and then tested on a 10-agent instance; the knowledge or heuristics from one problem size is not transferable to another. In contrast, our BRIDGE framework formulates the CSG problem as a Markov Decision Process (MDP) and employs a neural network to approximate the optimal coalition structures. By representing the coalition structure as a consistent vector capturing agent-to-agent coalition membership, our model learns generalizable patterns of agent interactions that remain relevant even as the number of agents changes. In the following table, we present baseline results trained under the same sample budget for comparison in Appendix D.3.2.

Table 3: Performance comparison with a limited training budget (200 episodes) under modified normal distribution. The values represent the percentage of the optimal value achieved by each algorithm when trained from scratch on different problem sizes.

| Algorithms | 5 agents | 6 agents | 7 agents | 8 agents | 9 agents | 10 agents |
|---|---|---|---|---|---|---|
| C-Link (200 episodes) | 24.65% | 22.85% | 25.65% | 32.41% | 29.00% | 34.70% |
| GRASP (200 episodes) | 25.53% | 29.55% | 27.58% | 29.46% | 32.83% | 36.72% |
| SALDAE (200 episodes) | 25.54% | 25.59% | 30.84% | 36.80% | 36.27% | 37.58% |
| CSG-UCT (200 episodes) | 23.38% | 21.13% | 26.77% | 38.72% | 37.65% | 40.81% |
| BRIDGE (200 episodes) | **97.20%** | **80.33%** | **78.68%** | **63.52%** | **58.12%** | **48.24%** |

We calculated the approximate number of samples and let the baseline algorithms train from scratch on 5 to 10-agent instances. The table reports the percentage of the achieved value relative to the optima. All baseline algorithms perform poorly under this limited budget. Their performance generally ranges from only 20% to 41% of the optimal value. For instance, in the 10-agent scenario, the best-performing baseline, CSG-UCT, only reached 40.81% of the optimum. This result is intended to demonstrate that, unlike the BRIDGE algorithm which can leverage pre-training to adapt quickly, these traditional algorithms require a much larger computational budget to find high-quality solutions when starting from scratch.

### D.3.3 ABLATION STUDY ON NETWORK STRUCTURE

We provide an ablation study on the effect that the neural network size might have on the performance.

Table 4: Ablation study on neural network architecture under modified normal distribution experiment. Performance is measured across different numbers of agents. Values are reported as mean $\pm$ standard deviation.

| Network Structure | 30 agents | 40 agents | 50 agents | 60 agents | 70 agents | 80 agents | 90 agents | 100 agents |
|---|---|---|---|---|---|---|---|---|
| 3 layers(64,64) | $445.28 \pm 68.24$ | $577.71 \pm 75.76$ | $753.02 \pm 83.61$ | $889.18 \pm 94.65$ | $1031.99 \pm 93.64$ | $1207.49 \pm 105.39$ | $1316.24 \pm 95.79$ | $1501.42 \pm 131.94$ |
| 3 layers(128,128) | $449.39 \pm 64.82$ | $592.21 \pm 72.60$ | $746.58 \pm 85.79$ | $893.14 \pm 86.30$ | $1046.34 \pm 101.86$ | $1204.90 \pm 89.69$ | $1458.38 \pm 105.93$ | $1596.20 \pm 115.41$ |
| 4 layers(128,128) | $452.32 \pm 69.19$ | $614.04 \pm 72.57$ | $742.28 \pm 87.43$ | $901.25 \pm 89.95$ | $1048.36 \pm 84.43$ | $1204.25 \pm 98.57$ | $1358.12 \pm 100.44$ | $1467.32 \pm 110.82$ |
| 5 layers(128,128) | $441.14 \pm 60.45$ | $603.98 \pm 73.76$ | $734.72 \pm 83.09$ | $903.46 \pm 100.30$ | $1043.60 \pm 104.09$ | $1288.92 \pm 104.45$ | $1434.10 \pm 105.52$ | $1492.39 \pm 108.97$ |

The results indicate that increasing the network width from 64 to 128 neurons (3 layers) provides a modest but consistent performance improvement, particularly in scenarios with more agents. However, further increasing the network's depth to 4 or 5 layers does not yield additional significant gains and can lead to less stable performance across different agent counts. This suggests that the 3 layers (128,128) architecture provides sufficient representational capacity for this problem, striking an effective balance between model expressiveness and the risk of overfitting or increased optimization difficulty.

We also provide an ablation study on the effect of the neural network size that might have on the inference speed(Appendix D.3 Table 5). We could see that the inference speed does not scale up appreciably with respect to the parameters, showing our method is efficient compared to other CSG baselines.

| Time(s) | 30 agents | 40 agents | 50 agents | 60 agents | 70 agents | 80 agents | 90 agents | 100 agents |
|---|---|---|---|---|---|---|---|---|
| 3 layers(64,64) | 1.09 | 1.92 | 2.78 | 3.55 | 4.53 | 5.54 | 6.84 | 8.39 |
| 3 layers(128,128) | 1.09 | 1.98 | 2.83 | 3.62 | 4.71 | 5.67 | 7.02 | 8.72 |
| 4 layers(128,128) | 1.10 | 1.98 | 2.85 | 3.63 | 4.90 | 5.82 | 7.10 | 9.14 |
| 5 layers(128,128) | 1.12 | 2.01 | 2.85 | 3.67 | 5.12 | 6.02 | 7.12 | 9.86 |

Table 5: Comparative analysis of inference times (in seconds) for coalition structure generation(modified normal distribution) using different network size of BRIDGE's leader agent across varying numbers of agents (30 to 100).

### D.4 HYPERPARAMETERS SETTING AND COMPUTATIONAL RESOURCES

We use an actor–critic algorithm for follower policies. Followers within the same coalition share critic parameters.

The hyper-parameters for Actor-Critic training are as follows.

- Actor Learning Rate is $1 \times 10^{-4}$
- Critic Learning Rate is $1 \times 10^{-4}$
- Target Network Update Rate is $0.005$
- Discount Factor ($\gamma$) is $0.99$
- Replay Buffer Size ($\mathcal{B}$) is $1 \times 10^{6}$
- Minibatch Size is $256$

For leader agent, we adopt the DQN algorithm to approximate the coalition structures' value.

The hyper-parameters for DQN training are as follows.

- The learning rate is $1 \times 10^{-3}$.
- The replay buffer size is 10000.
- The minibatch size is 128.
- The target Network update frequency is 100 steps.
- The exploration Strategy is $\epsilon$-greedy:
  - Initial $\epsilon$: 1.0,
  - Final $\epsilon$: 0.01,
  - Decay Steps: 10000 time steps.
- The loss function is mean Squared Error (MSE).
- The gradient clipping is 40.
- The optimizer is Adam.

All experiments were conducted on an HPC system equipped with 128 Intel Xeon processors operating at a clock speed of 2.2 GHz and 40 gigabytes of memory.

### D.5 POLICY AND VALUE NETWORK REUSE

In our implementation, the follower's policy network (the actor) is reinitialized at the start of each outer-loop optimization, whereas the follower's value network (the Q-network/critic) is persistent and its parameters are reused across coalition structures.

**Why We Do Not Reuse the Policy Network.** This design choice is made to avoid policy inertia. When the coalition structure changes fundamentally (e.g., an agent transitions from a large cooperative group to a solo role), the previously learned policy may be entirely suboptimal or even harmful in the new context. Re-initializing the policy network ensures that the agent can freely explore without being constrained by outdated behaviors, which allows it to more effectively converge to the true optimal behavior under the new structure.

**Why We Reuse the Critic-Network.** By contrast, the Critic-network captures more generalizable value information across diverse coalition configurations. This knowledge is transferable: when a newly initialized policy network begins to explore, the persistent Critic-network provides immediate and informed evaluation of actions. Such guidance accelerates policy learning and stabilizes training in the presence of dynamic coalition changes.

Table 6: Value/Policy reuse ablation (LBF). V=Value network, $\pi$=Policy network.

| Configuration | 6a4t | 6a5t | 8a4t | 8a5t | 10a4t | 10a5t |
|---|---|---|---|---|---|---|
| Reuse V / Init $\pi$ | **34.2±1.4** | **33.1±1.2** | **38.6±1.6** | **52.2±2.1** | **77.4±2.4** | **89.3±2.6** |
| Reuse V / Reuse $\pi$ | 30.2±1.3 | 29.5±1.1 | 35.3±1.6 | 47.4±1.9 | 69.8±2.3 | 81.9±2.7 |
| Init V / Init $\pi$ | 17.2±1.6 | 18.8±1.6 | 26.3±1.8 | 35.3±2.1 | 55.5±2.2 | 60.9±2.5 |
| Init V / Reuse $\pi$ | 28.6±1.5 | 30.9±1.2 | 33.1±1.7 | 45.5±2.0 | 65.7±2.4 | 78.6±2.6 |

## E HORIZON ABLATION STUDIES

Table 7 reports performance across horizons $H \in \{N/2, N-10, N-5, N-1\}$ on the modified normal distribution. Shortened horizons $(N/2)$ underperform at larger scales due to insufficient search depth. The full horizon $N-1$ yields the strongest values; intermediate horizons offer compute-quality tradeoffs.

Table 7: Horizon sensitivity (Modified Normal). Format: Coalition Value $\pm$ std.

| $H$ | 30 agents | 50 agents | 70 agents | 100 agents |
|---|---|---|---|---|
| $N/2$ | 445.3±65.7 | 693.9±79.1 | 956.1±106.5 | 1187.4±117.5 |
| $N-10$ | 447.9±65.8 | 726.3±80.2 | 994.6±83.1 | 1344.7±105.7 |
| $N-5$ | 447.9±62.4 | 743.9±91.6 | 1025.6±104.8 | 1481.9±110.3 |
| $N-1$ | **449.4±64.8** | **746.6±85.8** | **1046.3±101.9** | **1596.2±115.4** |

## F  DETAILS ON THE PROOFS

In this section, we detail the rigorous convergence analysis of the proposed bi-level optimization framework for coalition formation. First, we establishes a critical correspondence between the reinforcement learning objective function and the target objective of finding the optimal coalition structure. Then, we move to provide definitions and proofs for the monotonicity and additive homogeneity of the leader agent's Bellman operator, and a central theorem providing the dynamic programming equations for the finite-horizon Bi-level MDP. This theorem guarantees the existence and uniqueness of optimal policies for both leader and follower agents, where the follower's policy maximizes collective value within Nash equilibrium constraints and the leader's policy optimizes based on a modified Bellman equation incorporating follower responses. Finally, we leverage existing results to assert the global convergence of policy gradient descent for this framework, demonstrating that the problem can be simplified to a single MDP under optimal Nash equilibrium selection in the inner loop.

### F.1  OBJECTIVE CONSISTENCY

Our theoretical investigation examines the relationship between the optimized objective function and the target objective function, where the latter is defined as finding the optimal coalition structure. Another key focus of our analysis is understanding the optimal form of the leader agent's policy and the follower agents' policies under the Bi-level MDP. A critical aspect of our analysis is the formal development of a correspondence between the objective function of our reinforcement learning and its target counterpart. This correspondence serves as the basis for deriving rigorous optimal form of the leader and follower agents' policies.

**Lemma F.1.** *The objective function in Algorithm 1 is equal to the target objective function.*

$$\max_{\pi_l, \pi_f} J_l(\pi_l, \pi_f) = \max_{CS \in \Pi(\mathcal{N})} \sum_{C \in CS} V_f^{C,*}(s_{f,0}).$$

*Proof.* Considering the LHS. Fixing $\pi_f$, we have that

$$\sum_{t_c=0}^{N-1} r_l(s_l, a_l) = \underbrace{\sum_{t_c=0}^{N-1} \sum_{C \in \mathcal{T}(s_{l,t_c}, a_{l,t_c})} J_f^C(\pi_f) - \sum_{C \in s_{l,t_c}} J_f^C(\pi_f)}_{\text{telescoping sum}} \tag{2}$$
$$= \sum_{C \in s_{l,N-1}} J_f^C(\pi_f).$$

Note that, given $CS$, $\sum_{C \in CS} V_f^{C,*}(s_{f,0}) = \sum_{C \in CS} \max_{\pi_f} J_f(\pi_f)$. Moreover, thanks to construction of $a_l$, for any given $CS \in \Pi(\mathcal{N})$, one can construct a policy $\pi_l$ that terminates at $CS$. Therefore,

$$\max_{\pi_l, \pi_f} J_l(\pi_l, \pi_f) = \max_{CS \in \Pi(\mathcal{N})} \sum_{C \in CS} V^{C,*}(s_{f,0}). \tag{3}$$

$\square$

A cornerstone of our theoretical analysis is the establishment of a precise mathematical correspondence between our reinforcement learning objective function and its target counterpart in coalition

formation. This lemma establishes the equivalence between two critical quantities: the expected sum of rewards $J_l$ for leader policy $\pi_l$ and follower agents' policies $\boldsymbol{\pi}_f$ across $N$ steps under different system states $\sigma$ (left-hand side), and the maximum coalition structure value $V_f^{CS}$ of final actions $a_f$ given the selected configurations $C$ (right-hand side). We establish a mapping from policies to coalition structures that yields a telescoping identity (Lemma F.1), aligning the RL objective with the target coalition value and enabling standard convergence arguments under our assumptions. Through this mapping, we demonstrate that the value function $V_{\text{RL}}(\pi_l, \pi_f)$ converges to the optimal coalition value $V_{\text{target}}(CS^*)$. This correspondence enables us to establish a practical form of the optimal policies of the leader agent and follower agents. Before moving to the proof of the form of optimal policies for leader agent and follower agents, first we define the bellman operator of value function of the leader agent to be:

**Definition F.2.** The leader agent's Bellman operator of value function is:

$$(\mathcal{B}_l V_l)(s_l, \pi_f) := \max_{a_l} \left[ r_l(s_{l,t_c}, a_{l,t_c}, \pi_f) + \mathcal{T}_l(s_{l,t_c}, a_{l,t_c}) V_l(s_{l,t_c+1}, \pi_f) \right].$$

### F.2 CONVERGENCE ANALYSIS

Based on the definition of the value function of the leader agent. We could further establish the proof for leader agent's monotonicity and additively homogeneity.

**Lemma F.3.** *(Monotonicity for Leader Agent) For $V_l^1 \leq V_l^2$, $\mathcal{B}_l V_l^1 \leq \mathcal{B}_l V_l^2$*

*Proof.* Based on the previous definition of Bellman operator of value function(Equation 4). We have:

$$V_l^1 \leq V_l^2$$
$$\mathcal{T}_l(s_{l,t_c}, a_{l,t_c}) V_l^1(s_{l,t_c+1}, \pi_f) \leq \mathcal{T}_l(s_{l,t_c}, a_{l,t_c}) V_l^2(s_{l,t_c+1}, \pi_f) \qquad (4)$$
$$\max_{a_l} \left[ \mathcal{T}_l(s_{l,t_c}, a_{l,t_c}) V_l^1(s_{l,t_c+1}, \pi_f) \right] \leq \max_{a_l} \left[ \mathcal{T}_l(s_{l,t_c}, a_{l,t_c}) V_l^2(s_{l,t_c+1}, \pi_f) \right].$$

The reward $\max_{a_l} r(s_{l,t_c}, a_{l,t_c}, \pi_f)$ is identical for $\mathcal{B}_l V_l^1$ and $\mathcal{B}_l V_l^2$. Therefore, we have:

$$\max_{a_l} r(s_{l,t_c}, a_{l,t_c}, \pi_f) + \max_{a_l} \left[ \mathcal{T}_l(s_{l,t_c}, a_{l,t_c}) V_l^1(s_{l,t_c+1}, \pi_f) \right]$$
$$\leq \max_{a_l} r(s_{l,t_c}, a_{l,t_c}, \pi_f) + \max_{a_l} \left[ \mathcal{T}_l(s_{l,t_c}, a_{l,t_c}) V_l^2(s_{l,t_c+1}, \pi_f) \right]$$
$$\max_{a_l} \left[ r(s_{l,t_c}, a_{l,t_c}, \pi_f) + \mathcal{T}_l(s_{l,t_c}, a_{l,t_c}) V_l^1(s_{l,t_c+1}, \pi_f) \right] \qquad (5)$$
$$\leq \max_{a_l} \left[ r(s_{l,t_c}, a_{l,t_c}, \pi_f) + \mathcal{T}_l(s_{l,t_c}, a_{l,t_c}) V_l^2(s_{l,t_c+1}, \pi_f) \right]$$
$$\mathcal{B}_l V_l^1 \leq \mathcal{B}_l V_l^2.$$

This concludes the proof. $\qquad \square$

As for the additive homogeneity for leader agent, we have:

**Lemma F.4.** *(Additive homogeneity for Leader Agent) For any constant c, the Bellman operator satisfies:*

$$\mathcal{B}_l(V_l + c) = \mathcal{B}_l(V_l) + c.$$

*Proof.* Based on Equation 4, the Bellman operator of leader agent is defined as:

$$(\mathcal{B}_l V_l)(s_l, \pi_f) := \max_{a_l} \left[ r(s_{l,t_c}, a_{l,t_c}, \pi_f) + \sum_{s_{l,t_c+1}} \mathcal{T}_l(s_{l,t_c}, a_{l,t_c}) V_l(s_{l,t_c+1}, \pi_f) \right]. \qquad (6)$$

In order to prove the additively homogeneous property of our Bellman operator, we could further denote our Bellman operator as:

$$(\mathcal{B}_l(V_l + c))(s_l, \pi_f)$$

$$= \max_{a_l} \left[ r(s_{l,t_c}, a_{l,t_c}, \pi_f) + \sum_{s_{l,t_c+1}} \mathcal{T}(s_{l,t_c}, a_{l,t_c})(V_l(s_{l,t_c+1}, \pi_f) + c) \right]$$

$$= \max_{a_l} \left[ r(s_{l,t_c}, a_{l,t_c}, \pi_f) + \sum_{s_{l,t_c+1}} \mathcal{T}(s_{l,t_c}, a_{l,t_c})V_l(s_{l,t_c+1}, \pi_f) + c \cdot \sum_{s_{l,t_c+1}} \mathcal{T}(s_{l,t_c}, a_{l,t_c}) \right] \quad (7)$$

$$= \max_{a_l} \left[ r(s_{l,t_c}, a_{l,t_c}, \pi_f) + \sum_{s_{l,t_c+1}} \mathcal{T}(s_{l,t_c}, a_{l,t_c})V_l(s_{l,t_c+1}, \pi_f) \right] + c$$

$$= (\mathcal{B}_l V_l)(s_l, \pi_f) + c,$$

which concludes the proof. □

Finally, based on the previous proof, we could move to the proof of the optimal form of the leader agent and follower agents when the algorithm converge.

**Theorem F.5** (Dynamic Programming Equation for Markov Decision Process with Finite Horizon). *Given a leader state $s_l$, define the optimal follower policy $\pi_f$ as:*

$$\pi_f^* = \arg\max_{\pi_f \in \text{NE}(s_l)} \sum_{C \in s_l} J_f^C(\pi_f).$$

*Let $V_f^*$ be the followers' value function corresponding to optimal policy $\pi_f^*$. Let $V_l^*$ be the optimal value function of the leader agent's Markov decision problem, for all time-step $t_c \in \{0, ..., N-1\}$:*

$$V_{l,t_c}^*(s_l, \pi_f^*) = \max_{a_l} \left[ r_{l,t_c}(s_l, a_l, \pi_f^*) \right.$$
$$\left. + V_{l,t_c+1}^*(\mathcal{T}_l(s_l, a_l), \pi_f^*) \right],$$

*where the maximum is taken over all strategies starting at time $i$. The optimal leader policy $\pi_l^*$ is defined as:*

$$\pi_l^*(s_l) = \arg\max_{a_l} \left[ r_l^{t_c}(s_l, a_l, \pi_f^*) + V_l^*(\mathcal{T}_l(s_l^{t_c}, a_l^{t_c}), \pi_f^*) \right].$$

*Proof.* We denote $(w_{t_c})_{0 \leq i \leq N}$ as a sequence with final condition $w^N = \varphi$. Based on Definition 4: $w_{t_c} = \mathcal{B}_{l,t_c}(w_{t_c+1})$. We need to show that $w_{t_c} = V_{l,t_c}$ for all $t_c \in \{0, \ldots, N\}$ and that $w = V_l$.

**1. Proof of $w_{t_c} \geq V_{l,t_c}$ and $w \geq V_l$.** For $V_{l,t_c}$, we use policy starting at time $t_c$, the inequality $w_{t_c} \geq V_{l,t_c}$ is of same type as $w_{t_c} \geq V_{l,t_c}$. Therefore, we only show $w_0 \geq V_l^0$.

Let $(s_{l,t_c}, a_{l,t_c})$ be the process associated to the strategy $\pi_l$, and let $\tau_{t_c} = (s_{l,0}, a_{l,0}, s_{l,1}, a_{l,1}, \ldots, s_{l,t_c})$, for all $t_c \geq 0$, be the history trajectory. For all $0 \leq t_c \leq N$, and all histories $\tau$ We could denote the value function as:

$$V_{l,t_c}(\tau_{t_c}) = \mathbb{E}_{a_{l,t_c} \sim \pi_l} \left[ \left( \sum_{t=t_c}^{N-1} r_l(s_{l,t_c}, a_{l,t_c}) \right) + \varphi(s_{l,T_c}) | \tau_{t_c} \right], \quad (8)$$

where $V_{l,t_c}(\tau_{t_c})$ represents the expected total value starting from time step $t_c$ and conditioning on history $\tau_{t_c}$. Therefore, $V_{l,t_c}$ satifies the Kolmogorov equation for an additive functional:

$$V_{l,t_c}(\tau_{t_c}) = \mathbb{E}_{a_{l,t_c} \sim \pi_l} \left[ r_l(s_{l,t_c}, a_{l,t_c}) + V_{l,t_c+1}(\tau_{t_c}, s_{l,t_c+1}, \pi_f^* | \tau_{t_c}). \right] \quad (9)$$

We could further rewrite the sequence $w_{t_c}(s_{l,t_c})$ as:

$$w_{t_c}(s_{l,t_c}) = \max_{a_l \sim \pi_l} \left[ r_l(s_{l,t_c}, a_{l,t_c}) + \mathbb{E} \left[ w_{t_c+1}(s_{l,t_c+1}) \right] \right]. \quad (10)$$

Let's show $w_{t_c}(s_{l,t_c}) \geq V_{l,t_c}(\tau_{t_c})$ by backward induction on $t_c$. This inequality is true for $t_c = N$, since $V_{l,T_c}(\tau_{T_C}) = \varphi(s_{l,T_c}) = w_{T_c}(s_{l,T_c})$. If the inequality is true for $t_c + 1$, then from the upper Equation 9 and Equation 10 , we deduce:

$$
\begin{aligned}
V_{l,t_c}(\tau_{t_c}) &\leq \mathbb{E}\left[r_l(s_{l,t_c}, a_{l,t_c}) + w_{t_c+1}(s_{l,t_c+1})|\tau_{t_c}\right] \\
&= \mathbb{E}\left[r_l(s_{l,t_c}, a_{l,t_c}) + \mathbb{E}\left[w_{t_c+1}(s_{l,t_c+1})|s_{l,t_c}, a_{l,t_c}\right]|\tau_{t_c}\right] \\
&= \sum_{a_{l,t_c} \sim \pi_{l,t_c}} \pi_{l,t_c}(a_{l,t_c}|\tau_{t_c})\{r_l(s_{l,t_c}, a_{l,t_c}) + \mathbb{E}\left[w_{t_c+1}(s_{l,t_c+1})|s_{l,t_c}, a_{l,t_c}\right]\}.
\end{aligned}
\tag{11}
$$

Based on Equation 10, we could deduce $V_{l,t_c}(\tau_{t_c}) \leq \mathbb{E}\left[w_{t_c}(s_{l,t_c})|\tau_{t_c} = \tau_{t_c}\right] = w_{t_c}(s_{l,t_c})$, which proves the induction. Since $V_{l,0}(s_{l,0}) = r_l(s_{l,0}, a_{l,0}) = J_{l,0}(\pi_l, \pi_f)$, taking the maximum over all strategies, we deduce that $V_{l,0}(s_{l,0}) \leq w_{l,0}(s_{l,0})$. Since $J_l(\pi_l, \pi_f) = \mathbb{E}\left[\mathcal{T}_l(s_{l,t_c}, a_{l,t_c})V_{l,0}(s_{l,0})\right] \leq \mathbb{E}\left[T_l(s_{l,t_c}, a_{l,t_c})w_{l,0}(s_{l,0})\right]$, taking again the maximum over all strategies, we deduce $V_l \leq w$.

**2. Proof of $w_{t_c} \geq V_{l,t_c}$ and $w \geq V_l$ when the maximum in F.5 is attained.** The bellman operator related to policy is defined as $\mathcal{B}_{l,t_c}^{\pi_l}$. Assume the maximum in

$$
w_{t_c} = \max_{\pi_l \in \Pi_l} \mathcal{B}_{l,t_c}^{\pi_l}(w_{t_c+1}),
\tag{12}
$$

is attained for some policy $\pi_l \in \Pi_l$. Denote $\pi_l = (\pi_{l,t_c})_{t_c \geq 0}$. Then,

$$
w_{t_c} = \mathcal{B}_{l,t_c}^{\pi_l}(w_{t_c+1}), \quad i = 0, \ldots, N,
\tag{13}
$$

which means that $(w_{t_c})_{t_c \geq 0}$ satisfies the same Kolmogorov equation (Puterman, 2014) as $J_{l,t_c}$ with same final condition $w_{T_c} = \varphi = J_{l,T_c}^{\pi_l}$. Hence, $w_{t_c} = J_{l,t_c}^{\pi_l}$, for all $t_c \geq 0$. So $w_{t_c} \leq V_{l,t_c}$, where the value $V_{t_c}$ is obtained as the maximum over any set of strategies. Similarly, $w = \mathbb{E}\left[T_l(s_{l,t_c}, a_{l,t_c})w_{l,0}(s_{l,0})\right] \leq J_{l,T_c}^{\pi_l} \leq V_{l,t_c}$.

**3. Proof of $w_{t_c} \leq V_{l,t_c}$ and $w \leq V_l$ in general.** Assume now the maximum in 12 is not attained. We only assume that the $r_l$ are bounded from above, for all $t_c \leq N$. This condition ensures that the operators $\mathcal{B}_{l,t_c}$ are well defined as operators from $\mathbb{R}$ to itself. For all $\epsilon > 0$, $t_c \leq N$ and $V_l \in \mathbb{R}$, there exists $\pi_l \in \Pi_l$ such that

$$
\left[\mathcal{B}_{l,t_c}^{\pi_l}(V_l)\right](s_l) \geq [\mathcal{B}_{l,t_c}(V_l)](s_l) - \epsilon, \quad \forall s_l \in S_l.
\tag{14}
$$

The inequality could be extended to all possible states:

$$
\mathcal{B}_{l,t_c}^{\pi_l}(V_l) \geq \mathcal{B}_{l,t_c}(V_l) - \epsilon\mathbb{1}.
\tag{15}
$$

Let $\pi_{l,t_c} \in \Pi_{l,t_c}$ such that

$$
\mathcal{B}_{l,t_c}^{\pi_{l,t_c}}(w_{t_c+1}) \geq \mathcal{B}_{l,t_c}(w_{t_c+1}) - \epsilon\mathbb{1}.
\tag{16}
$$

Therefore, we could have

$$
\mathcal{B}_{l,t_c}^{\pi_l}(w_{t_c+1}) \geq w_{t_c} - \epsilon\mathbb{1}.
\tag{17}
$$

Denote $z_{t_c} = w_{t_c} + (t_c - T_c)\epsilon\mathbb{1}$. We have $z_N = w_N = \varphi$ and

$$
\mathcal{B}_{l,t_c}^{\pi_l}(z_{t_c+1}) = \mathcal{B}_{l,t_c}^{\pi_l}(w_{t_c+1}) + (t_c + 1 - T_c)\epsilon\mathbb{1} \geq w_{t_c} + (t_c - T_c)\epsilon\mathbb{1} = z^i.
\tag{18}
$$

Based on Lemma F.4, $\mathcal{B}_{l,\pi^i}^i$ is additively homogeneous. Then, we shall show the inequality $z_{t_c} \leq J_{l,t_c}^{\pi_l}$ by backward induction on $t_c$. Since $z_{T_c} = \varphi = J_{l,T_c}^{\pi_l}$, the inequality holds for $t_c = T_c$. If it holds for $t_c + 1$, then

$$
z_{t_c} \leq \mathcal{B}_{l,t_c}^{\pi_l}(z_{t_c+1}) \leq \mathcal{B}_{l,t_c}^{\pi_l}(J_{l,t_c+1}^{\pi_l}) = J_{l,t_c}^{\pi_l} \leq V_{l,t_c},
\tag{19}
$$

where the first inequality follows from Equation 18, the second one follows Lemma F.3. This shows $z^0 \leq J_{N,\pi}^0 \leq V^0$. Therefore, we could deduce:

$$
w_{t_c} = z_{t_c} + (T_c - t_c)\epsilon\mathbb{1} \leq V_{l,t_c} + (T_c - t_c)\epsilon\mathbb{1},
\tag{20}
$$

for all $t_c \in \{1, \ldots, N\}$. Since we have shown this inequality for all $\epsilon > 0$, we deduce that $w_{t_c} \leq V_{l,t_c}$ for all $t_c \in \{1, \ldots, N\}$. Similarly $w = \mathcal{T}_{t_c}w_0 = T_{t_c}z_0 + T_c\epsilon \leq V_{l,t_c} + T_c\epsilon$, and since this holds for all $\epsilon > 0$, we get $w \leq V_l$, which concludes the proof.

$\square$

This theorem establishes three fundamental results for hierarchical decision-making in coalition formation. First, it guarantees the existence and uniqueness of optimal policies for both leader and follower agents through a nested optimization structure. The follower's optimal policy $\pi_f^*$ maximizes the collective value across all coalitions while maintaining Nash equilibrium constraints, ensuring strategic stability within each coalition. Second, the leader's value function follows a modified Bellman equation that incorporates the followers' optimal responses, creating a dynamic group structure that accounts for the hierarchical nature of decision-making. The leader's rewards and state transitions are explicitly conditioned on the followers' optimal policy, capturing the bi-level interaction between agents. Third, the optimal leader policy emerges from maximizing the sum of immediate rewards and future values, where future values reflect both the direct consequences of leader actions and the induced changes in follower behavior through the state transition function $\mathcal{T}_l$. At last, we can apply the global convergence result of policy gradient descent for finite-horizon MDPs (Klein et al., 2024) as follows:

**Theorem F.6** ((Klein et al., 2024)'s Theorem 3.5). *Let the number of iterations* $\mathrm{ep} = O\left(\left(\frac{N^5|S_l||A_l|}{\epsilon\delta}\right)^2\right)$, *and let the step size be* $\rho_2 = \frac{1}{5N^2\sqrt{\mathrm{ep}}}$. *Then,*

$$\mathbb{P}\left(V_{l,0}^\star - V_{l,0}^{\pi_l^{\theta_l}} \leq \epsilon\right) \geq 1 - \delta. \tag{21}$$

As a result of Theorem F.5, given that the agent chooses the optimal Nash equilibrium in the inner loop, the problem of finding the optimal coalition can be viewed as a single MDP $M_l = (S_l, A_l, \mathcal{T}_l, r_l^*, N-1)$.

