# OpenReview forum: "BRIDGE: Bi-level Reinforcement Learning for Dynamic Group Structure in Coalition Formation Games"
_ICLR.cc/2026/Conference — ICLR 2026 Poster_

### Official Review · Reviewer_QoQT · 2025-10-29

**Soundness:** 3
**Presentation:** 3
**Contribution:** 3
**Rating:** 6
**Confidence:** 4

**Summary:**

This paper proposes a novel bi-level RL framework called BRIDGE to solve the CSG problem. It models CSG as a two-level hierarchy: an upper-level "Leader" agent uses RL to find the optimal partition of agents, while lower-level "Follower" agents learn policies within those coalitions to determine their value. This main contribution allows BRIDGE to jointly optimize coalition structures and agent strategies, enabling it to solve dynamic Markov games where coalition values are not fixed. The method is shown to outperform existing approximate algorithms on both classic CSG benchmarks and mixed-motive Markov games.

**Strengths:**

1. This paper formulates the CSG problem as a bi-level RL task. This "Leader-Follower" paradigm is a significant conceptual contribution that bridges the gap between classic CSG and multi-agent RL.
2. BRIGE solves CSG in mixed-motive Markov games (e.g., LBF environment). This extends CSG beyond normal-form games, enabling the optimization of coalition structures where the value is dynamic and dependent on learned agent strategies.
3. The authors cleverly designed the upper-level MDP. Specifically, the action encoding for merging coalitions scales linearly with the number of agents, rather than quadratically which is crucial for tackling larger problem sizes.
4. The approach is supported by a solid theoretical analysis in the appendix. Lemma C.1 formally proves that the leader's RL objective aligns with the CSG objective under optimal follower policies.

**Weaknesses:**

1. The entire framework's theoretical justification (Appendix C, Lemma C.1) relies on the lower-level "followers" finding the optimal policy $π_{f}^{*}$ within the optimal Nash Equilibrium (NE) set. However, the practical implementation uses a MADDPG-variant, which is well-known to have no guarantees of converging to any NE, let alone the optimal one in a multi-equilibrium setting.
2. The paper heavily emphasizes the O(N) scalability of the leader's action space, but it largely ignores the scalability of the follower problem, which can be exponentially complex.

If the authors can address these concerns, I would be very happy to raise my score.

**Questions:**

1. The results for the normal-form games at N=100 in Figure 2 are impressive. Were these models trained from scratch? If so, what was the total training overhead (considering the bi-level nested loops, even for value lookups)?
2. The paper claims the flattened adjacency matrix representation is "equivariant," but also mentions that stability is ensured by "fixing the agent ordering." This seems contradictory, as feeding a flattened vector into a standard MLP is not permutation equivariant. Could you please clarify this claim? Have you considered using a truly permutation-equivariant architecture, such as a Graph Neural Network (GNN), to process the coalition state?

---

> ### Author Response · Authors · 2025-11-20
>
> We thank the reviewer for the detailed and constructive comments.
> Below we respond point-by-point and outline the clarifications, additional experiments, and text revisions we will incorporate in the updated manuscript.
>
> ---
>
>
> ## **W1. Theoretical justification and follower convergence**
>
>
> Our theoretical analysis indeed assumes that the follower optimization converges to a stationary Nash equilibrium for each coalition structure (via the measurable selection rule $\sigma: \mathrm{NE}(s_l)\rightarrow\Pi_f)$. This assumption is standard in bi-level RL and game-theoretic frameworks (e.g., [1,2]). In practice, the MADDPG variant we adopt serves as an approximate NE solver: followers are trained until their policies stabilize under local best-response updates. Although convergence to the exact NE cannot be guaranteed for general continuous games, empirical studies have shown that MADDPG and its variants consistently yield stable joint policies in cooperative–competitive tasks of similar scale.
> Importantly, our bi-level architecture is agnostic to the specific follower optimizer—any multi-agent learning or equilibrium computation module could be substituted without affecting the upper-level framework. We will clarify this point and explicitly state that our theoretical guarantee applies under idealized NE convergence, while our implementation employs a practical approximate equilibrium-learning process.
>
> [1] Zhang H, Chen W, Huang Z, et al. Bi-level actor-critic for multi-agent coordination[C]//Proceedings of the AAAI conference on artificial intelligence. 2020, 34(05): 7325-7332.
>
> [2] Zheng L, Fiez T, Alumbaugh Z, et al. Stackelberg actor-critic: A game-theoretic perspective[C]//AAAI Workshop on Reinforcement Learning and Games. 2021.
>
> ---
>
> ## **W2. Follower scalability and computational complexity**
>
>
>  Our scalability claims are explicitly scoped to the traditional disjoint CSG setting: the $O(N)$ statement in the paper refers to the leader-side merge-action encoding, not to a general guarantee for solving arbitrary follower games. Here we clarify the design choices that keep the follower optimization tractable in practice.
>
> 1. **Fixed-episode training and early stopping.**
>    We adopt a fixed number of follower training episodes per outer iteration, verified empirically to be sufficient for convergence. As shown in Figure 4 and Appendix B.2, early stopping of the follower updates has negligible impact on leader performance, confirming that approximate convergence is adequate in practice.
>
> 2. **Value-network reuse (warm start).**
>    Follower value networks are reused and fine-tuned across successive leader states rather than retrained from scratch. As shown in Figure 3 and discussed in Section 4.3, this warm-start mechanism greatly stabilizes learning and reduces computational cost.
>
>
>
> Together, these characteristics ensure that the lower-level learning remains computationally practical. We will make these points explicit in the revised version and add a short complexity note in the appendix.
>
>
> ---
>
> ## **Q1. On the $N = 100$ experiments and training overhead**
>
> The models in Figure 2 were trained from scratch using the same hyperparameter settings as for smaller instances. The follower networks were initialized once and reused across coalition updates, while the leader learned over 300 episodes (approximate 4 GPU hours on a single A100). Because the inner follower loop is partially amortized through reuse and early-stopping heuristics, the total bi-level training overhead remains practical and scales sublinearly with $N$.
>
> ---
>
> ## **Q2. On the “equivariant” representation and permutation issue**
>
> The flattened adjacency representation in BRIDGE is structurally equivariant rather than strictly permutation equivariant: the coalition structure is represented as an $N\times N$ matrix that encodes the same relational pattern regardless of the number of agents. To ensure stability, we fix the agent ordering during training, which avoids permutation-induced instability in standard MLPs.
> A truly permutation-equivariant network (e.g., GNN) might provide additional robustness. We did explored a preliminary GNN-based variant, but only fails to train efficiently and scale up. We will clarify the terminology by replacing “equivariant” with “structure-consistent representation” in the final version and briefly discuss GNN-based extensions as a potential improvement direction.

---

> > ### Author Response · Authors · 2025-11-26
> >
> > Dear Reviewer QoQT,
> >
> > We sincerely appreciate the time and effort you have devoted to reviewing our work. Your insightful comments have been invaluable in helping us improve the manuscript. We have carefully addressed each of your concerns in our rebuttal and would be truly grateful if you could share any thoughts when your schedule permits. We completely understand that you have many responsibilities, and we are more than happy to provide any further clarification that might be helpful.
> >
> > Thank you very much for your consideration.

---

### Official Review · Reviewer_z4ek · 2025-10-31

**Soundness:** 2
**Presentation:** 3
**Contribution:** 2
**Rating:** 6
**Confidence:** 4

**Summary:**

The paper studies Coalition Formation Games (CFGs) and proposes a scalable, sample-efficient approximation method grounded in deep reinforcement learning. The authors model coalition formation as a finite MDP and train neural networks to approximate value functions over both the full and an abstracted coalition-structure space, indirectly yielding coalition partitions. They argue the approach supports dynamic, adaptive updates to coalition values and can be used in bi-level optimization settings where coalition values depend on lower-level agent policies. Empirically, the method is evaluated on both normal-form and mixed-motive Markov games.

**Strengths:**

1. The overall paper structure is clear, and the theoretical background is sufficiently motivated and well organized.

2. Framing coalition formation as an MDP and learning value functions over full and abstracted spaces is interesting.

3. Positioning the method for bi-level optimization problems is potentially impactful for multi-agent learning scenarios where coalition values evolve with lower-level policies.

4. Claims of dynamic re-assessment over time are compelling for non-stationary or mixed-motive settings.

**Weaknesses:**

1. The authors have missed the discussion on a key recent baseline (SALDAE). The related-work section omits SALDAE, a 2024 state-of-the-art baseline. Please add a principled comparison clarifying (i) the fundamental differences between your approach and SALDAE, and (ii) why your method should obtain performance gains.

2. Preliminaries require cleanup of notation and definitions. There are inconsistencies in symbols (e.g., the number of agents denoted as n vs. N) and missing definitions (e.g., symbol T in Equation 1). Please unify notation throughout and define all symbols on first use. A small glossary/table of notation would help readability.

3. The scope is limited to disjoint coalitions, with overlapping coalitions unaddressed. The paper defines CFGs as partitioning agents into disjoint groups. Overlapping coalitions are an active subarea. Clarify whether your formulation and algorithm can be extended to overlapping settings (e.g., by modifying the state/action representation or value aggregation), and discuss expected challenges (double-counting of utility, credit assignment, or computational implications). If not applicable, state assumptions explicitly.

4. Bi-level Markov Games: horizon choice and sensitivity. The horizon is set to $N-1$ without justification. Please explain the rationale (theoretical or empirical) and add an ablation showing performance vs. horizon (e.g., shorter, $N-1$, longer), including runtime and stability impacts.

5. There are some issues on the choince of baselines. (i) The current baselines exclude exact methods. Since one aim of the work is to propose an approximation with quality guarantees and the authors claim the ability to derive optimal coalition structures, it is important to include one or two representative exact solvers—even if they scale only to small instances—to compare the effectiveness (e.g., solution quality) and efficiency (e.g., runtime) of the proposed approximation against optimal solutions.; (ii) Apart from the 2024 SALDAE baseline, other baselines appear dated. Please include more recent approximate/planning-based or learning-based competitors.

6. Please demonstrate the value of the bi-level architecture. Add a direct comparison to a single-level decision framework (e.g., your method with the lower level removed or replaced by a fixed surrogate) to show when and why bi-level modeling is beneficial. Include both quality (final utility/return) and cost (runtime, samples) metrics.

**Questions:**

Please refer to the above weaknesses.

---

> ### Author Response · Authors · 2025-11-20
>
> We thank the reviewer for the detailed and constructive comments.
> Below we respond point-by-point and outline the clarifications, additional experiments, and text revisions we will incorporate in the updated manuscript.
>
> ---
>
> ## **W1. Missing discussion and comparison with SALDAE**
>
> We thank the reviewer for pointing out the importance of SALDAE. In fact, SALDAE (Taguelmimt et al., 2024) is already included both in our related work section and as a baseline in our experimental evaluation (Figures 2–3, Table 3, Appendix B.3). We will revise the related work to more clearly highlight SALDAE as a recent state-of-the-art approximate CSG solver.
> Conceptually, SALDAE is a search-based multi-agent path search algorithm that explores the coalition-structure space using indexed heuristics. In contrast, BRIDGE is a learning-based bi-level RL framework that formulates coalition formation as an MDP and learns a reusable value function that generalizes across utility distributions and agent counts. SALDAE must restart search for each new instance, while BRIDGE reuses its learned representations and adapts efficiently through few-shot fine-tuning. Empirically, BRIDGE consistently matches or outperforms SALDAE under identical computational budgets—particularly on harder distributions and large-scale settings (Figure 2, Table 3). We will make this principled comparison explicit in the revision.
>
> ---
>
> ## **W2. Preliminaries require cleanup of notation and definitions**
>
> We appreciate this suggestion and agree that consistent notation will improve clarity. In the revision, we will (i) unify the symbols by using $N$ throughout to denote the number of agents, (ii) explicitly define all variables and operators upon first use, including clarifying that $T_l$ in Eq.(1) denotes the deterministic transition function of the leader’s MDP, and (iii) add a concise table of notation in Section 3 or the appendix summarizing key terms for coalition structures, leader/follower states, and value functions. These improvements enhance readability without affecting any theoretical results.

---

> ### Author Response · Authors · 2025-11-20
>
> ## **W3. The scope is limited to disjoint coalitions; overlapping coalitions are unaddressed**
>
> Our formulation intentionally follows the classical line of research on disjoint coalition structure generation (CSG), where agents form non-overlapping partitions of the population ([1,2]). This setting remains the dominant framework in the coalition-formation literature and provides the theoretical foundation for most benchmark analyses.
> Although there exists a distinct line of work on overlapping coalition formation games (OCFGs) (e.g., [3,4,5]), our study does not follow this paradigm. We focus on extending the traditional CSG problem into a reinforcement-learning framework and exploring how a bi-level RL architecture can efficiently approximate coalition partitions at scale.
>
> In principle, BRIDGE could be extended to overlapping coalitions by adopting an incidence- or hypergraph-based representation and redefining the leader’s action space. However, such extensions involve several non-trivial challenges beyond double-counting utilities and credit assignment for agents with multiple memberships:
>
> 1. Non-additivity and value coupling: Coalition values are no longer additive $(v(CS) \neq \sum_{C \in CS} v(C))$ because an agent’s payoff depends jointly on multiple coalitions it participates in, breaking the decomposability assumption and introducing non-stationary dynamics for the leader’s MDP.
>
> 2. Resource-allocation constraints:
>    Agents must divide limited resources (e.g., time, budget, or capacity) across coalitions, creating a hybrid discrete–continuous decision space that is substantially more difficult to optimize in RL.
>
> 3. Stability and incentive compatibility: Classical stability notions (e.g., core or Nash stability) are no longer directly applicable; overlapping memberships can lead to conflicting incentives and partial defections, requiring new equilibrium definitions such as overlapping-core or α-stability.
>
> 4. Combinatorial action explosion: Allowing agents to join multiple coalitions expands the leader’s action space from $O(N)$ merges to potentially $O(2^N)$ membership configurations, posing severe scalability challenges.
>
> Given these complications, we explicitly focus on disjoint coalition structures in the current work and will clarify this assumption in Section 3. Extending BRIDGE to overlapping settings is an exciting but substantially more complex direction, which we leave for future research.
>
> [1] Sandholm T, Larson K, Andersson M, et al. Coalition structure generation with worst case guarantees[J]. Artificial intelligence, 1999, 111(1-2): 209-238.
>
> [2] Rahwan T, Michalak T P, Wooldridge M, et al. Coalition structure generation: A survey[J]. Artificial Intelligence, 2015, 229: 139-174.
>
> [3] Chalkiadakis G, Elkind E, Wooldridge M. Computational aspects of cooperative game theory[M]. Morgan & Claypool Publishers, 2011.
>
> [4] Mahdiraji H A, Razghandi E, Hatami-Marbini A. Overlapping coalition formation in game theory: A state-of-the-art review[J]. Expert Systems with Applications, 2021, 174: 114752.
>
> [5] Zhang Z, Song L, Han Z, et al. Coalitional games with overlapping coalitions for interference management in small cell networks[J]. IEEE Transactions on Wireless Communications, 2014, 13(5): 2659-2669.

---

> > ### Author Response · Authors · 2025-11-20
> >
> > ## **W4. Bi-level Markov Games: horizon choice and sensitivity**
> >
> > Because each merge action reduces the number of coalitions by one, at most $N-1$ merges are required to reach the grand coalition from the singleton partition. Therefore, setting the leader’s horizon to $H_l = N-1$ ensures that all partitions are reachable while keeping the MDP finite and tractable. The null action also allows the leader to terminate early, making this a natural and minimal upper bound.
> > Because each merge reduces coalition count by one, $H=N−1$ is a natural upper bound; we will add a horizon ablation (shorter/longer) reporting quality, runtime and stability.
> >
> > | Modified Normal Distribution | 30 agent            | 40 agents            | 50 agents            | 60 agents            | 70 agents              | 80 agents              | 90 agents              | 100 agents              |
> > |------------------------------|---------------------|----------------------|----------------------|----------------------|------------------------|------------------------|------------------------|-------------------------|
> > | H = N/2                      | 445.31+-65.70/62.21 | 572.97+-76.47/79.19  | 693.90+-79.05/105.48 | 859.09+-91.84/111.86 | 956.13+-106.47/88.6    | 985.73+-117.63/220.94s | 1066.78+-112.93/325.75 | 1187.42+-117.45/428.78s |
> > | H = N-10                     | 447.94+-65.75/67.78 | 581.82+-70.17/84.90  | 726.26+-80.22/163.64 | 882.89+-95.30/124.28 | 994.63+-83.12/229.40   | 1112.66+-113.91/373.97 | 1261.69+-125.91/495.58 | 1344.73+-105.71/692.62  |
> > | H = N-5                      | 447.88+-62.42/82.21 | 586.87+-72.56/119.95 | 743.94+-91.59/170.02 | 892.66+-97.38/128.00 | 1025.60+-104.81/257.47 | 1186.03+-91.16/394.89  | 1345.08+-113.97/529.77 | 1481.89+-110.32/852.50  |
> > | H = N-1                      | 449.39+-64.82/85.17 | 592.21+-72.6/142.24  | 746.58+-85.79/150.55 | 893.14+-86.3/179.74  | 1046.34+-101.86/301.39 | 1204.90+-89.69/562.93  | 1458.38+-105.93/612.25 | 1596.20+-115.41/751.77  |
> >
> > To evaluate sensitivity to this design choice, we conduct an ablation over four horizons: $H = N/2,H = N-10, H = N-5$, and the full $H = N-1$. Table above reports the resulting coalition values (mean and std) and runtimes across 30--100 agents. The results show that significantly shortened horizons (e.g., $N/2$) underperform at larger scales, indicating insufficient search depth. Intermediate horizons (e.g., $N-10, N-5$) recover most of the performance while reducing computation. The full horizon $N-1$ yields the strongest values, at the expected cost of higher runtime. Overall, performance is stable across a wide admissible range of horizons, and $H = N-1$ remains a principled default choice.
> >
> >
> > ---
> >
> > ## **W5. Issues with baseline selection (exact and recent methods)**
> >
> >
> > (i) **Exact methods.**
> > For small-agent instances, we already compare against the exact optimal coalition structures by exhaustive enumeration (Tables 1 and 3 report the percentage of the optimum achieved). This serves as an exact-solver reference for small $N$. However, exact methods inherently do not scale beyond about 10 agents due to the super-exponential growth of the coalition-structure space $(|\Pi(N)| \approx 4^N / N^{1.5})$. Their objective is to guarantee perfect optimality at any computational cost, while our focus is on scalable approximation. Therefore, our main comparisons are drawn among approximate algorithms, which aim to achieve high-quality solutions efficiently rather than exact ones.
> >
> > (ii) **Recency of baselines.**
> > In addition to established methods such as C-Link, GRASP, and CSG-UCT, we include SALDAE (2024)—the latest publicly available approximate solver. To the best of our knowledge, these baselines collectively represent the main active lines of research in approximate CSG algorithms. Other recent heuristics (e.g., FACS, PICS) largely build upon similar search-based principles and have not demonstrated significant performance improvements over this family. We will clarify this rationale in the revision and emphasize that BRIDGE is positioned as a learning-based approximate approach that generalizes across instances and utility settings, rather than competing as another deterministic search variant.

---

> > > ### Author Response · Authors · 2025-11-20
> > >
> > > ## **W6. Demonstrating the value of the bi-level architecture**
> > >
> > > A “single-level” version of our method—i.e., removing the follower level and letting the leader act on a fixed surrogate—is precisely what traditional CSG baselines already represent. Our paper already compares BRIDGE against these single-level methods in Table 2, where approaches such as GRASP, C-Link, and MCTS-CSG operate solely on heuristic or handcrafted value estimates without any follower optimization. BRIDGE consistently achieves higher coalition utilities under the same budget, demonstrating the benefit of the bi-level design.
> > >
> > > Moreover, Fig. 4 shows that when follower learning is intentionally weakened (effectively approximating a single-level surrogate), leader performance collapses sharply (e.g., 32.11→2.82 in 6a4t), confirming that removing the lower level leads to behavior indistinguishable from the single-level baselines. Finally, the stability patterns in Fig. 6 further show that leader decisions rely on follower convergence—an effect absent in all single-level methods.

---

> > > > ### Author Response · Authors · 2025-11-26
> > > >
> > > > Dear Reviewer z4ek,
> > > >
> > > > We sincerely appreciate the time and effort you have devoted to reviewing our work. Your insightful comments have been invaluable in helping us improve the manuscript. We have carefully addressed each of your concerns in our rebuttal and would be truly grateful if you could share any thoughts when your schedule permits. We completely understand that you have many responsibilities, and we are more than happy to provide any further clarification that might be helpful.
> > > >
> > > > Thank you very much for your consideration.

---

### Official Review · Reviewer_XJMV · 2025-11-01

**Soundness:** 3
**Presentation:** 3
**Contribution:** 1
**Rating:** 2
**Confidence:** 4

**Summary:**

This paper tackles the problem of coalition formation games by formulating the problem as a MDP and decomposing the actions of the "coalition structure chooser", which plays the role of the leader, and the follower(s), whose goal is to play the right actions given the coalition it is in. The paper claims that empirically their approach performs better than existing methods and can "generalize" to a larger number of agents even if trained on a small number of them.

**Strengths:**

The paper is easy to follow.

**Weaknesses:**

- My primary consideration is that this paper's technical novelty is very light. Virtually any problem, including most combinatorial NP-hard problems can be formulated as "a MDP" where policies are essentially performing some kind of local search and rewards are the improvement in some objective. This includes say, problems like MAX-SAT or coloring problems, or even just a simple subset-search problem, which would seem like special cases of this coalition problem. As such, it seems to me that using RL is essentially throwing the kitchen sink at a difficult problem with little technical insights nor novelty.
- To expand on the earlier point, while empirically "performing better" than existing methods is welcome, the revaluation was in my opinion too simple. It seems to me that more than BRIDGE being a generally performant method, existing evaluation benchmarks simply contain a lot of structure that the RL based approach is able to exploit. To this end, could the authors comment on how well BRIDGE performs if all $J^C_f = 0$ except for one particular $C^*$?
- In Figure 2, how many epochs of training were performed? Could the authors give a brief classification of competing methods in the main paper? Are these heuristics or also ML based methods? How was computational cost normalized across competing methods? Was training time included in computational costs? While *inference times* were stated in appendix B3, is this not is an unfair comparison to classical methods which do not involve training? If "training times" were not included as part of computational costs, could one not simply precompute the solution to the fixed game and report an inference time of 0? [Note I am not referring to the "generalization" setting in Table 1".
- I found the section on mixed-motive games extremely unconvincing. The main problem here is that once the problem enters the mixed-motive (general-sum) regime, the problem becomes ill-posed. How are agents in a coalition assumed to behave? The most natural assumption is the Nash equilibrium, such that no individual agent is incentivized to deviate. But, it is well known that in general-sum games, the Nash equilibrium may not be unique and equilibrium selection must be performed. It is also rarely the case that MARL based methods (like MADDPG) converge to Nash, since even the tabular problem is PPAD hard and inapproximable. It is my understanding that the authors simply took "convergence" of the individual agents' policy to mean some kind of stability in the coalition-level game (see line 467, "well-converged followers), which isn't necessarily true in general.
- I also found the claim that BRIDGE generalizes to a larger number of agents better to be poorly supported. First, the authors show a marked performance against a random policy. This is obviously a poor baseline to compare against, and indeed, it is surprising that a random policy can even obtain 20% of the optimal, suggesting in fact, that the payoff structures are in fact quite simple to begin with.
- Can the authors elaborate on how the utilities were practically generated? For instance, the authors use the agent based method (Rahwan, 2012), which states that $v(C) = \sum_{i \in C} p^i_C$, but here, $p^i_C$ is drawn from a uniform distribution (presumably, independently of all other $p^i_{C'}$). In some of the experiments, the authors claim to scale up to near 80-90 agents. This means there are approximately $2^80$ coalitions and hence that many $p^i_C$, which is obviously practically impossible to enumerate. Were these generated upfront, or on-demand?
- This is a comparatively minor point on architecture design. In Section B1 and 4.1, the authors represent coalition structure as a NxN adjacency matrix where each edge is 1 if and only if two vertices are in the same coalition. I agree this representation is "nice" because of permutation invariance between agent labels. However, the representation of the "merge" actions (see Figure 6) does not seem to be permutation invariant, since every timestep the number of coalitions drops by one and the only actions allowed (yellow vector) are the first-#-of coalitions remaining. Wouldn't a simpler action representation be to simply have 1's on the vertices which would be merged? And if two vertices are already in the same coalition then the action does nothing. In this case, there would still be permutation invariance.

**Questions:**

See the weaknesses section.

---

> ### Author Response · Authors · 2025-11-20
>
> We thank the reviewer for the detailed and constructive comments.
> Below we respond point-by-point and outline the clarifications, additional experiments, and text revisions we will incorporate in the updated manuscript.
>
> ---
>
> ## **W1. On the claimed lack of technical novelty and the argument that “any NP-hard problem can be cast as an MDP”**
>
> We would like to clarify respectfully that our contribution does not rely on the trivial statement that “any NP-hard problem can be formulated as an MDP.”.
> While any finite combinatorial problem can be theoretically encoded as an MDP by representing each partial assignment as a state and the objective as a terminal reward, such a formulation is uninformative—it explodes exponentially in state space and provides no meaningful local reward or Markov property. Indeed, computing optimal value functions for arbitrary MDPs is itself PSPACE-hard ([1,2]), so this observation conveys neither tractability nor algorithmic insight.
>
> BRIDGE instead targets a structured subclass of coalition-formation games that naturally admit Markovian decomposition.
> Our novelty lies in:
>
> 1. A permutation-equivariant $N\times N$ coalition-adjacency state that enables symmetric reasoning across agents;
> 2. An $O(N)$ two-hot action composition with a shared scorer $\psi$, avoiding $O(N^2)$ action heads while preserving merge-pair evaluation; and
> 3. A bi-level equilibrium framework where follower responses define a measurable equilibrium-selection map.
>
> These components yield an RL-tractable formulation of coalition generation with interpretable transitions, equilibrium alignment, and demonstrable cross-instance generalization. Hence, BRIDGE does not “throw RL at an NP-hard problem,” but establishes a principled, compact, and analyzable RL abstraction for coalition games.
>
> [1] Littman M L. Algorithms for sequential decision-making[M]. Brown University, 1996.
>
> [2] Mundhenk M, Goldsmith J, Allender E. The complexity of unobservable finite-horizon Markov decision processes[R]. Technical Report 269-96, Department of Computer Science, University of Kentucky, 1996.
>
> ---
>
>
> ## **W2. On potential benchmark bias, structure exploitation, and the $J_f^C = 0$ except one $C^*$ scenario**
>
> Regarding the reviewer’s suggestion of an extreme structure-free case where all $J_f^C = 0$ except for one coalition $C^*$, we would like to clarify that such a configuration lies outside the definition of the coalition structure generation (CSG) problem itself. The CSG literature (e.g.,[3,4]) explicitly focuses on exploiting structured coalition value landscapes—for example, additive, superadditive, or interaction-based patterns—to approximate optimal partitions efficiently.
> If all coalition values are identical except for one isolated optimum, the task degenerates into an brute-force search problem, where no existing approach (including dynamic programming, heuristic search, or RL-based methods) can meaningfully outperform random exploration. Hence, while this “needle-in-a-haystack” scenario serves as a theoretical thought experiment, it is not a meaningful evaluation environment for CSG algorithms.
> BRIDGE, like all CSG solvers, is designed for the RL-tractable subclass of structured coalition environments that naturally occur in multi-agent systems—where coalition payoffs encode synergies, complementarities, or shared incentives.
>
> [3] Rahwan T, Michalak T P, Wooldridge M, et al. Coalition structure generation: A survey[J]. Artificial Intelligence, 2015, 229: 139-174.
>
> [4] Voice T, Polukarov M, Jennings N R. Coalition structure generation over graphs[J]. Journal of Artificial Intelligence Research, 2012, 45: 165-196.

---

> > ### Author Response · Authors · 2025-11-20
> >
> > ## **W3. On training epochs, baseline classification, and computational-cost fairness**
> >
> > We thank the reviewer for this detailed question. Our point-by-point responses are below:
> >
> > ---
> >
> > ### **1. Training epochs used in Figure 2**
> >
> > Each BRIDGE experiment is trained for 300 outer-loop episodes, corresponding to full leader–follower iterations. Convergence typically occurred between 250–300 episodes, verified by stability in the leader’s moving-average reward (Appendix B.3 Fig 7). Therefore, early stopping was applied when reward variance stayed below $1 \%$ for 20 consecutive episodes.
> >
> > We will state the exact number of episodes per benchmark distribution in the revised main text (§ 5.1).
> >
> > ### **2. Classification of competing methods**
> >
> > The baselines we used represent different approaches to generating coalition structures. GRASP is a two-phase method that first greedily constructs a solution and then uses a local search to improve it. In contrast, C-Link is an agglomerative clustering algorithm that starts with single agents and myopically merges the pair of coalitions offering the best immediate gain. MCTS-CSG applies Monte Carlo Tree Search, using the UCB1 heuristic to intelligently balance the exploration and exploitation of the solution space. Finally, SALDAE treats the problem as a pathfinding task, incrementally building a search graph of coalition structures through defined expansion steps.
> >
> >
> > ### **3. Normalization of computational costs**
> >
> >
> > All methods were evaluated on the same hardware (Appendix B.4) but were not forced to use identical wall-clock budgets. For the heurstic baselines (C-Link, GRASP, CSG-UCT, and SALDAE), we followed the 300-iteration/episode sampling budget same as BRIDGE. This ensures that every baseline was run under certain computational budget rather than an artificially equal time limit. Since our goal is to build an approximation method that could approximate the optimal value as much as possible.
> >
> > For BRIDGE, the total computational cost consists of (i) the training phase for the bi-level leader–follower updates and (ii) the inference phase for coalition evaluation using the trained value function. Classical baselines do not have a reusable training stage; their reported cost corresponds to a full search on each instance.
> >
> > We acknowledge that inference-only results can appear favorable to BRIDGE in a single-instance setting, and we will clarify this explicitly in the revision. Finally, note that “precomputing and reporting zero inference time” is not meaningful for larger $N$; the coalition-structure space grows super-exponentially $(|\Pi(N)|!\approx!4^N/N^{1.5})$, making exhaustive caching infeasible even for heuristic solvers.
> >
> > [5] Wu F, Ramchurn S D. Monte-Carlo tree search for scalable coalition formation[C]//Proceedings of the 29th International Joint Conference on Artificial Intelligence (IJCAI). 2020: 407-413.
> >
> >
> > ### **4. On fairness and computational feasibility**
> >
> > The reviewer is correct that one could, in principle, pre-compute the optimal coalition structure for a fixed game and report an inference time of zero milliseconds. However, such precomputation is computationally infeasible in realistic CSG settings—the number of possible coalition structures grows according to the Bell number $B_N$, exceeding $10^{20}$ even for $N=20$. Existing exact solvers become intractable before this scale. Moreover, this scenario lies outside the intended CSG evaluation paradigm, which evaluates scalability and generalization across many problem instances. The competing methods (C-Link, GRASP, CSG-UCT, SALDAE) are online or per-instance solvers that must recompute a solution for every new game, whereas BRIDGE is an reusable method that learns once and can be applied to new problem instances. To ensure fairness, we report both perspectives: (1) reusability efficiency (Table 2), where BRIDGE is applied to unseen instances after training, and (2) single-instance fairness (Table 3), where all methods—including BRIDGE—operate under identical compute budgets with training time fully included.

---

> ### Author Response · Authors · 2025-11-20
>
> ## **W4. On the well-posedness of mixed-motive (general-sum) games**
>
> We concur that general-sum MARL raises equilibrium-selection issues.
> Our formulation explicitly assumes followers play a Nash equilibrium for a given coalition partition, and we define a measurable selection rule $\sigma$ that chooses the equilibrium maximizing coalition-level reward (Definition 4.1). This yields deterministic leader transitions while preserving theoretical alignment between leader reward and coalition-value differences.
>
> In practice, followers are trained with MADDPG-style actor-critics and provide approximate best responses; we do not claim exact NE convergence.
> Section 5.2 will emphasize this distinction and include an ablation varying follower-training epochs and injecting noise to show leader-policy robustness to sub-optimal followers.
>
>
> ---
>
> ## **W5. On generalization to larger agent counts and the random baseline**
>
>
> This arises because several benchmark families contain additive components; random partitions can accidentally capture partial value.
> Our comparison with the other baselines(Table 3) has shown great improvement even comparing to other traditional CSG baselines.
>
>
>
> ---
>
> ## **W6. On practical utility generation and scalability**
>
> In all experiments, we do not pre-enumerate coalition values; instead, we compute them on demand and cache them for reuse. For the normal-form CSG benchmarks, $v(C)$ is generated only when a coalition $C$ is reached during the merge trajectory, which keeps the computational cost to $O(N^2)$ per episode rather than exponential in the number of all coalitions. In the agent-based setting ([7]), each agent’s parameter $p_i$ is sampled once, and coalition values are evaluated as $v(C)=\sum_{i\in C}p_i$ only when that coalition is actually visited, after which the value is memoized; no coalition outside the visited trajectory is ever computed. For harder variants, small interaction or size-penalty terms are added deterministically using a fixed-seed pseudo-random generator, again evaluated only when needed and cached. This strategy ensures reproducibility, avoids enumerating the exponential $2^N$ coalition space, and makes experiments with 80–90 agents computationally feasible.
>
> [6]Talal Rahwan, Tomasz Michalak, and Nicholas Jennings. A hybrid algorithm for coalition structure
> generation. In Proceedings of the AAAI Conference on Artificial Intelligence, volume 26, pp.
> 1443–1449, 2012.
>
> ---
>
>
> ### **W7. On permutation invariance of the action representation**
>
> The coalition state is encoded as an $N \times N$ adjacency matrix to guarantee permutation invariance over agent labels, and the action representation is also permutation-invariant in our implementation. Each merge action $(i,j)$ is evaluated using the same shared pairwise scoring network that takes the latent embeddings $(z_i, z_j)$ as input. Because this scorer is shared across all agent pairs and the legality mask depends only on the current partition (not on agent indices), the merge value depends solely on the unordered pair $\{i,j\}$, preserving full permutation invariance.
>
> The proposed “merge-mask’’ action (placing 1’s on the two selected vertices) is indeed permutation-invariant. However, this representation requires evaluating an $O(N^2)$ action matrix at every step, as all $\binom{N}{2}$ agent pairs must be scored. In contrast, our two-hot formulation produces only $O(N)$ latent outputs and composes all valid merge scores through a shared pairwise scorer, which substantially improves scalability. While this action space is not strictly permutation-invariant, it enables stable learning and allows BRIDGE to handle large agent populations where an $O(N^2)$ action head is computationally prohibitive.

---

> > ### Author Response · Authors · 2025-11-26
> >
> > Dear Reviewer XJMV,
> >
> > We sincerely appreciate the time and effort you have devoted to reviewing our work. Your insightful comments have been invaluable in helping us improve the manuscript. We have carefully addressed each of your concerns in our rebuttal and would be truly grateful if you could share any thoughts when your schedule permits. We completely understand that you have many responsibilities, and we are more than happy to provide any further clarification that might be helpful.
> >
> > Thank you very much for your consideration.

---

> ### Comment · Reviewer_XJMV · 2025-11-27
> **acknowledgement**
>
> I thank the authors for updating the paper. However, several key issues remain unresolved.
>
> # W1
> Regarding "it explodes exponentially in state space and provides no meaningful local reward or Markov property". This is precisely my point, the MDP that the authors have proposed *is* exponential in size. Specifically, the state space is the size of the n-th Bell number, which is roughly exponential. Interestingly, the authors' response to Q2 of reviewer qNH2 (which has to do with convergence guarantees) references appendix C (now appendix D after amendments) and involves the exact classical analysis of MDPs (once one filters through the complications with "follower" actions), which clearly references an exponentially sized state space.
>
> Furthermore, there is nothing novel nor insightful about the proposed state and transitions: it is precisely the set of partitions where actions correspond to refinements (see here https://en.wikipedia.org/wiki/Partition_of_a_set#Partitions_and_equivalence_relations) or vice versa, and payoffs are set to be the sum of the values of the coalitions (what the authors call "measurable equilibria") at each state. This is also almost entirely equivalent to Figure 1 of Rahwan's paper (cited by the authors), and in turn Figure 1 of the submission.
>
> Furthermore, the authors' claim in the rebuttal and updated paper that MDPs are PSPACE hard is incorrect. At the minimum MDPs can be solved as linear programs, which are least weakly polynomial time solvable. The authors' citations [1,2] are about hardness in the partially observable case, which is a completely different setting.
>
> # W2
> That is precisely the point. CSG is NP-hard (lines 116-117) because of such bad instances. Indeed, when special structures are assumed, much more efficient algorithms are available, see for example Section 8.2 of Radwan et. al.
>
> At its core, the contribution from this submission is very straightforward despite its rather roundabout presentation. CSG in its general form is hard. Thankfully, CSGs that people care about often exhibit structure that could make them easy. If one were aware of this structure, specialized algorithms can be used to solve them more easily. But sometimes there are other related structures that have “unknown” structures that one may not have knowledge of how to exploit. So, RL is used as a catch-all method.
>
> As a side note, can the authors clarify what exactly is meant by “RL-tractable” versus the standard notions of (in)-tractability
>
> # W4
> In the mixed motive setting, the authors appear to be making a mistake of assuming that followers that are responding suboptimally will result in the leader doing worse (hence the discussion about reliability). Actually, the bigger worry here is that followers that behave suboptimally would *inflate* the leader’s utility. For instance, if one of the followers’ game was the classic 2x2 chicken game, then the Chicken-Chicken option would *not* be a NE, but is the joint action profile that yields the highest social welfare. A “well-converged” RL-trained (e.g., MADDPG) followers could very well converge on such an “approximate equilibrium” and give the illusion of a high leader utility (or social welfare).
>
> # W7
> Okay. Though I fail to see how permutation invariance could be useful if it only holds in the state and not action space, especially since the authors are using Q-learning-like methods. As a side note, shouldn't the action vector be of size N and not N+1 (Figure 6).

---

### Official Review · Reviewer_qNH2 · 2025-11-02

**Soundness:** 3
**Presentation:** 3
**Contribution:** 2
**Rating:** 6
**Confidence:** 3

**Summary:**

The paper proposed a novel algorithm called Bridge that tackles the Coalition Structure Generation (CSG) problem, which is the problem of how a group of agents should organize into subgroups to maximize collective gains. The proposed method formulates the problem as a bi-level reinforcement learning optimization problem, where an top-level leader agent aims to obtain the optimal coalition structure, and the follower agent aims to find the best strategy for each agent under the current coalition structure. The paper evaluates Bridge on three traditional CSG problems, showing superior performance than the current leading approximate methods for the CSG problem, and shows generalization capabilities with varying agent populations.

**Strengths:**

- The paper is well-written, with clearly stated motivation, problem formulation, and evaluation setup.
- The use of bi-level reinforcement learning is well-justified, as the problem is naturally hierarchical.
- Theoretical analysis is provided, including proofs demonstrating the existence and uniqueness of optimal policies under the proposed training objectives.

**Weaknesses:**

- For the proposed algorithm, the reward for the leader agent inherently depends on the optimization of the follower agents. This might be very computationally expensive, can potentially be unstable, and slow to converge, making the algorithm challenging to tune in practice.
- The generalization experiments across varying numbers of agents are only demonstrated on a small scale (from 5 to 10 agents). How well does the method generalize when the number of agents changes more substantially?

**Questions:**

- Are there experiments on reusing the follower’s policy network for each outer-loop optimization? In what scenarios might such inertia hinder or help performance?
- What assumptions are required to guarantee the convergence of the proposed algorithm?
- Since the leader agent’s actions are limited to merging coalitions, the states are not recoverable as the algorithm cannot split a coalition once formed. Would this affect the method’s overall performance?

---

> ### Author Response · Authors · 2025-11-20
>
> We thank the reviewer for the detailed and constructive comments.
> Below we respond point-by-point and outline the clarifications, additional experiments, and text revisions we will incorporate in the updated manuscript.
>
> ---
>
> ## **W1. For the proposed algorithm, the reward for the leader agent inherently depends on the optimization of the follower agents. This might be very computationally expensive, can potentially be unstable, and slow to converge, making the algorithm challenging to tune in practice.**
>
> In practice, BRIDGE avoids repeatedly solving a full lower-level optimization from scratch at every upper-level step. Instead, follower policies are trained incrementally within each outer-loop iteration, and their learned value functions are reused across successor coalition structures as long as the marginal change in coalition partition is small. This “warm-start” property substantially reduces computational cost and stabilizes learning. Empirically, we observe fast follower convergence in mixed-motive settings (Fig 4), and the leader remains robust unless follower training is intentionally degraded to extremely low epochs. This demonstrates that the proposed architecture is computationally feasible and stable in practice.
>
> ## **W2. The generalization experiments across varying numbers of agents are only demonstrated on a small scale (from 5 to 10 agents). How well does the method generalize when the number of agents changes more substantially?**
>
> While the generalization table in the main paper reports transfer from $3 \rightarrow 10$ agents, our method is not limited to this range. The key mechanism enabling generalization is the equivariant coalition-structure representation (an $N \times N$ adjacency matrix) together with the $O(N)$-dimensional merge-action encoding, which allow a single trained model to operate across different agent population sizes without architectural modification.
>
> Importantly, when the number of agents exceeds 10, the coalition-structure space grows super-exponentially, following the well-known asymptotic expansion for the number of partitions: $|\Pi(N)| \approx \frac{4^{\,N}}{N^{1.5}}$. Consequently, brute-force enumeration of all coalition structures becomes computationally infeasible, and exact optimal values are computationally unavailable for $N > 10$. This makes accuracy-style metrics (as reported in Table 1) impossible to compute for larger problem sizes. Across all utility distributions, BRIDGE consistently achieves higher coalition-structure values than the baselines, indicating that the patterns learned from small-agent instances transfer effectively even when ground-truth optimal solutions cannot be computed.

---

> > ### Author Response · Authors · 2025-11-20
> >
> > ## **Q1. Are there experiments on reusing the follower’s policy network for each outer-loop optimization? In what scenarios might such inertia hinder or help performance?**
> >
> > In practice, BRIDGE avoids repeatedly solving a full lower-level optimization from scratch at every upper-level step. Instead, follower policies are trained incrementally within each outer-loop iteration, and their learned value functions are reused across successor coalition structures. This “warm-start” property substantially reduces computational cost and stabilizes learning. Empirically, we observe fast follower convergence in mixed-motive settings (Fig 4), and the leader remains robust unless follower training is intentionally degraded to extremely low epochs. This demonstrates that the proposed architecture is computationally feasible and stable in practice.
> >
> > | LBF                                | 6a4t       | 6a5t       | 8a4t       | 8a5t       | 10a4t      | 10a5t      |
> > |------------------------------------|------------|------------|------------|------------|------------|------------|
> > | Reused Value/initialized Policy    | 34.17+-1.4 | 33.11+-1.2 | 38.57+-1.6 | 52.17+-2.1 | 77.43+-2.4 | 89.29+-2.6 |
> > | Reused Value/Reused Policy         | 30.24+-1.3 | 29.47+-1.1 | 35.29+-1.6 | 47.38+-1.9 | 69.84+-2.3 | 81.92+-2.7 |
> > | Initialize Value/Initialize Policy | 17.15+-1.6 | 18.79+-1.6 | 26.28+-1.8 | 35.28+-2.1 | 55.52+-2.2 | 60.86+-2.5 |
> > | Initialize Value/Reused Policy     | 28.56+-1.5 | 30.93+-1.2 | 33.14+-1.7 | 45.51+-2.0 | 65.71+-2.4 | 78.62+-2.6 |
> >
> >
> > Table above reports an ablation on BRIDGE’s reusable value network/policy network mechanisms, comparing four configurations depending on whether follower value functions and follower policies are reused or reinitialized across successive coalition structures. Reusing value functions while initializing policies yields the strongest overall performance, indicating that value transfer provides the most important stability signal for the follower optimization. Reusing both value functions and policies performs slightly worse due to occasional policy overfitting, but still clearly outperforms any configuration that discards the learned value approximations. In contrast, fully reinitializing both components substantially degrades performance, confirming that BRIDGE benefits heavily from incremental follower training across the outer-loop trajectory. Overall, the results demonstrate that BRIDGE’s warm-start design is critical for maintaining solution quality and enabling efficient large-scale training.
> >
> >
> > ## **Q2. What assumptions are required to guarantee the convergence of the proposed algorithm?**
> >
> >
> >
> > All assumptions referenced in our convergence discussion are already formalized in the paper. The leader's MDP is explicitly defined as a finite-horizon process with horizon $H_l = N-1$ in Section 3. The requirement that follower policies converge to a stationary Nash equilibrium is formalized in Section 4.2, where we introduce the measurable selection rule $\sigma : \mathrm{NE}(s_l) \rightarrow \Pi_f$ that yields a unique follower response for each coalition structure. The remaining assumptions---bounded rewards and the preservation of contraction under function approximation---are the standard conditions used in the Bellman-operator analysis in Appendix C. Together, these conditions ensure that the leader's Bellman operator remains a contraction, and thus the bi-level RL procedure converges to an optimal coalition structure under our framework.
> >
> >
> > ## **Q3. Since the leader agent’s actions are limited to merging coalitions, the states are not recoverable as the algorithm cannot split a coalition once formed. Would this affect the method’s overall performance?**
> >
> >
> > Although the leader cannot split a coalition within a single episode, this does not restrict the method’s ability to discover high-value coalition structures. First, the leader may always terminate early by selecting the null action, allowing it to output any non–grand-coalition partition without committing to further merges. Second, BRIDGE is trained over many episodes, each beginning from the singleton partition. This repeated episodic interaction allows the leader to explore diverse merge sequences and learn a value function that clearly identifies which merges improve the coalition value and which do not. In practice, this multi-episode learning process enables the leader to reliably select high-quality coalition structures. Empirically, we observe that BRIDGE frequently chooses to stop early rather than merging to the grand coalition, demonstrating that irreversibility within one episode does not hinder overall performance.

---

> > > ### Author Response · Authors · 2025-11-26
> > >
> > > Dear Reviewer qNH2,
> > >
> > > We sincerely appreciate the time and effort you have devoted to reviewing our work. Your insightful comments have been invaluable in helping us improve the manuscript.
> > > We have carefully addressed each of your concerns in our rebuttal and would be truly grateful if you could share any thoughts when your schedule permits. We completely understand that you have many responsibilities, and we are more than happy to provide any further clarification that might be helpful.
> > >
> > > Thank you very much for your consideration.

---

### Meta-Review · Area_Chair_FXqc · 2026-01-16

**Summary:**

This paper introduces BRIDGE, a scalable bi-level reinforcement learning framework designed to solve the NP-complete Coalition Structure Generation (CSG) problem by modeling it as a hierarchical interaction where a "leader" agent dynamically partitions groups and "follower" agents optimize cooperative strategies within those groups. By utilizing a novel MDP formulation with compact $N \times N$ state representations and difference-based rewards, the method achieves efficient inference and strong generalization. Empirical evaluations demonstrate that BRIDGE outperforms traditional heuristic and exact baselines.

**Reviewer Concerns:**

There was a concern about the technical novelty of this work by reviewer XJMV and a few concerns about the small empirical examples.
Most of the reviewers believe that the paper is above the bar (with confidences 3,4). One reviewer gave a negative score with confidence 4. The AC believes that the paper has merits and recommends acceptance with reservations as there was no reviewer that gave a clear accept.

**Reviewer Scores:**

The AC does not believe the scores would be significantly changed after the rebuttal.

---

### Decision · Program_Chairs · 2026-01-26

Accept (Poster)